# Restraint of presynaptic protein levels by Wnd/DLK signaling mediates synaptic defects associated with the kinesin-3 motor Unc-104

Jiaxing Li[1†], Yao V Zhang[2,3†‡], Elham Asghari Adib[1], Doychin T Stanchev[2,3], Xin Xiong[1], Susan Klinedinst[1], Pushpanjali Soppina[1], Thomas Robert Jahn[4§], Richard I Hume[1], Tobias M Rasse[2,4#*], Catherine A Collins[1*]

[1]Department of Molecular, Cellular, and Developmental Biology, University of Michigan, Ann Arbor, United States; [2]Junior Research Group Synaptic Plasticity, Hertie-Institute for Clinical Brain Research, University of Tübingen, Tübingen, Germany; [3]Graduate School of Cellular and Molecular Neuroscience, University of Tübingen, Tübingen, Germany; [4]CHS Research Group Proteostasis in Neurodegenerative Disease, DKFZ Deutsches Krebsforschungszentrum, Heidelberg, Germany

**\*For correspondence:**
tobias.rasse@googlemail.com (TMR);
collinca@umich.edu (CAC)

[†]These authors contributed equally to this work

**Present address:** [‡]The Picower Institute for Learning and Memory, Department of Biology and Department of Brain and Cognitive Sciences, Massachusetts Institute of Technology, Cambridge, United States; [§]AbbVie Deutschland GmbH, Ludwigshafen, Germany; [#]Advanced Light Microscopy Facility, European Laboratory of Molecular Biology, Heidelberg, Germany

**Competing interests:** The authors declare that no competing interests exist.

**Abstract** The kinesin-3 family member Unc-104/KIF1A is required for axonal transport of many presynaptic components to synapses, and mutation of this gene results in synaptic dysfunction in mice, flies and worms. Our studies at the *Drosophila* neuromuscular junction indicate that many synaptic defects in *unc-104-null* mutants are mediated independently of Unc-104's transport function, via the Wallenda (Wnd)/DLK MAP kinase axonal damage signaling pathway. Wnd signaling becomes activated when Unc-104's function is disrupted, and leads to impairment of synaptic structure and function by restraining the expression level of active zone (AZ) and synaptic vesicle (SV) components. This action concomitantly suppresses the buildup of synaptic proteins in neuronal cell bodies, hence may play an adaptive role to stresses that impair axonal transport. Wnd signaling also becomes activated when pre-synaptic proteins are over-expressed, suggesting the existence of a feedback circuit to match synaptic protein levels to the transport capacity of the axon.
DOI: https://doi.org/10.7554/eLife.24271.001

## Introduction

Synapse development, maintenance and plasticity involve highly orchestrated trafficking events in both pre and postsynaptic cells. In contrast to postsynaptic receptors, whose trafficking and organization has been studied extensively in many different synapse types (*Choquet and Triller, 2013*), much less is known about the mechanisms that regulate the assembly and maintenance of the neurotransmitter release machinery in the presynaptic neuron. This machinery includes the active zone (AZ), an electron-dense complex of structural proteins that scaffold both calcium channels and synaptic vesicles (SV) for the coordination of calcium-regulated exocytosis (*Südhof, 2012*). The protein components of the AZ are synthesized in cell bodies and trafficked together in association with vesicles (known as piccolo-bassoon transport vesicles (PTVs) (*Ahmari et al., 2000*; *Maas et al., 2012*; *Shapira et al., 2003*). SV precursors are also synthesized in cell bodies, and carried by kinesin motors to synapses (*Hall and Hedgecock, 1991*; *Okada et al., 1995*). Regulation of synapse development likely involves a global coordination of the synthesis and transport of both AZ and SV

**eLife digest** Each nerve cell, or neuron, has a long nerve fiber – called an axon – that forms specialized sites for information exchange – called synapses – with other cells. Many molecules work at synapses to coordinate the exchange of information. These molecules are largely made in the central part of the neuron – known as the cell body – and are then transported along the axon to the synapses.

The transport of these molecules is carried out by proteins known as molecular motors. One molecular motor, called KIF1A in humans and Unc-104 in fruit flies, is thought to be a major transporter of synaptic molecules. Mutations that hinder this molecular motor result in neurons failing to form synapses and, instead, synaptic components accumulate in the cell body. However, it was not clearif Unc-104 does actually carry all of the components needed to assemble synapses along axons, or if it influences synapse formation in another way.

Now, Li, Zhang et al. report new evidence that supports the second of these two hypotheses. The experiments made use of fruit flies in which the gene for Unc-104 had been deleted, and revealed that inhibiting enzymes in a specific signaling pathway could reverse the synaptic problems caused by the loss of Unc-104. The signaling pathway, which is conserved between flies and humans, involves an enzyme that is called Wnd in flies and DLK in humans. The Wnd/DLK signaling pathway was previously known to regulate how neurons respond when their axons are damaged (either by growing new axons or dying, depending on the context).

Further investigation by Li, Zhang et al. revealed that signaling via the Wnd enzyme becomes triggered whenever the Unc-104 molecular motor is impaired. This activation correlates with the build-up of synaptic proteins in the cell body. Once activated, the pathway then reduces the total amount of synaptic proteins that the cell makes. This reduction matches the neuron's reduced ability to transport them along the axon, and may help the neuron to adapt when axonal transport is impaired. However, the reduction in synaptic proteins also impaired the exchange of information at the synapses.

These findings suggest how DLK could be behind problems with synapses in diseases in which transport along axons is impaired. These diseases include hereditary spastic paraplegia, which has been linked to mutations in human KIF1A, and may also include ALS and Alzheimer's disease, which have recently been linked to DLK.

DLK has received recent attention as a candidate drug target because it contributes to the deterioration of damaged neurons. These new findings further expand that interest by suggesting that inhibiting DLK may help neurons to maintain working synapses, which is more useful than simply preventing damaged neurons from dying.

DOI: https://doi.org/10.7554/eLife.24271.002

components. However the mechanisms that regulate these important steps in synapse development and maintenance are poorly understood.

A critical role in synapse development has been assigned to the kinesin-3 family of motor proteins (*Hall and Hedgecock, 1991*; *Kern et al., 2013*; *Niwa et al., 2016*; *Pack-Chung et al., 2007*; *Yonekawa et al., 1998*). Mutations in mammalian *Kif1a* and its *unc-104* orthologues in *C. elegans* and *Drosophila* (also known *as imac, Klp53D and bris in Drosophila*) cause severe defects in synapse development. In *Drosophila unc-104-null* mutants, synaptic boutons fail to form, SV and AZ components fail to traffic to nascent synapses, and concomitantly, SV and AZ associated proteins accumulate in the cell body (*Pack-Chung et al., 2007*). It is broadly accepted that Unc-104 protein functions as a molecular motor to physically deliver presynaptic components to their destinations in the synaptic terminal (*Goldstein et al., 2008*). However, while there is biochemical evidence that KIF1A can interact with and 'carry' SV precursors (*Okada et al., 1995*), there is little evidence that KIF1A (or Unc-104) carries AZ components. The mechanistic role of Unc-104 in AZ transport and assembly remains unclear.

In this study we found that synaptic defects in embryonic *unc-104-null* mutants, including the failure to form synaptic boutons and AZs, arise not from direct loss of Unc-104 transport function, but via an indirect mechanism, which involves activation of the Wnd/DLK axonal damage signaling

pathway. The Wnd/DLK mixed lineage kinase has recently received intense interest for its roles in regulating both regenerative and degenerative responses to axonal damage in vertebrate and invertebrate neurons (*Gerdts et al., 2016*; *Hao and Collins, 2017*; *Li and Collins, 2017*; *Tedeschi and Bradke, 2013*). We found that the Wnd/DLK signaling pathway becomes activated when Unc-104's function is impaired, and then promotes synaptic dysfunction by restraining expression of multiple pre-synaptic AZ and SV protein components. This restraint concomitantly reduces protein buildup in cell bodies, which may play an adaptive role to stresses that disrupt intracellular transport, and contribute to pathologies that arise when transport is disrupted.

## Results

### Roles for the Unc-104 kinesin in AZ transport and synaptic bouton growth can be functionally separated from SV transport

Previous studies of NMJ development in *unc-104-null* mutant animals have revealed essential roles for this kinesin in synaptic maturation (*Hall and Hedgecock, 1991*; *Kern et al., 2013*; *Pack-Chung et al., 2007*; *Yonekawa et al., 1998*). At the *Drosophila* neuromuscular junction (NMJ), *unc-104-null* mutants are severely defective in the formation of presynaptic boutons, fail to localize SVs to NMJ terminals and show strong reductions in AZ localization (*Pack-Chung et al., 2007* and *Figure 1*). We found that disrupting the axonal damage signaling kinase Wnd (*wnd³* single mutants) had no significant effect on bouton formation, AZ number, or presynaptic protein localization, but double mutants with *unc-104-null* alleles (*unc-104^{P350}*, *unc-104^{170}*, and *unc-104^{52}*), gave a very informative phenotype: in *unc-104^{null};wnd^{3/3}* double mutants the synaptic bouton formation (*Figure 1A and C*), AZ number (*Figure 1A and D*), and synaptic levels of the AZ protein Brp (*Figure 1A and E*) were restored to a wild type phenotype. In contrast, the synaptic levels of SV proteins (VGlut, SytI and CSP) remained negligible, as in the *unc-104^{null}* single mutants (*Figure 1B and E*). These results are consistent with previous findings that Unc-104 functions as an essential molecular motor to transport SV precursors to synaptic terminals. However our findings indicate that transport of AZ precursors and bouton formation can occur independently of Unc-104. These defects are mediated by a second and separable mechanism, which depends upon the function of the Wnd kinase.

### The Wnd signaling pathway mediates presynaptic defects in *unc-104-hypomorph* mutants

To further study the effect of Wnd upon AZ assembly we utilized several hypomorphic *loss-of*-function mutations of *unc-104*, whose ability to survive to the third instar larval stage has allowed for extensive characterization of synaptic defects associated with Unc-104 (*Barkus et al., 2008*; *Cao et al., 2014*; *Kern et al., 2013*; *Zhang et al., 2016*; *2017*). We used the EMS-generated alleles *bris/null* (*Kern et al., 2013*) and *O3.1/null* (*Barkus et al., 2008*), and knockdown in motoneurons via independent RNAi lines. All of these mutants share similar characteristics: Post Synaptic Densities (PSDs) form (identified by the presence of a core receptor subunit GluRIII), however ~50% lack apposing presynaptic AZ components, as identified by Brp (*Figure 2A and C*), Liprin-α (*Figure 2B and D*) and the voltage gated calcium channel Cac-GFP (*Figure 2—figure supplement 2*). This defect is accompanied by a reduction in the levels of Brp within individual synapses and also across entire NMJ terminals (*Figure 2E*). Similarly to the *unc-104-null* mutants (*Figure 1*), mutations in *wnd* fully rescued these presynaptic AZ assembly defects (*Figure 2A–E*). RNAi knockdown of *unc-104* using either pan-motoneuron or single motoneuron driver lines led to AZ defects and concomitant knockdown of *wnd* in neurons led to rescue, (*Figure 2—figure supplement 1*), indicating a cell-autonomous role for both Unc-104 and Wnd in the synaptic defects.

In axonal regeneration and synaptic overgrowth, Wnd has been shown to act in a signaling pathway consisting of a cascade of MAP kinases and transcription factors (*Collins et al., 2006*; *Nakata et al., 2005*; *Xiong et al., 2010*). We found that inhibition of the downstream MAP Kinase JNK (Bsk) and Fos transcription factor, via expression of dominant-negative (DN) isoforms in neurons, could also rescue the presynaptic defect shown in the *unc-104* single mutants (*Figure 2C* and *Figure 2—figure supplement 2*). These results suggest that a signaling pathway consisting of Wnd, JNK and Fos mediates synaptic defects observed in *unc-104* mutants.

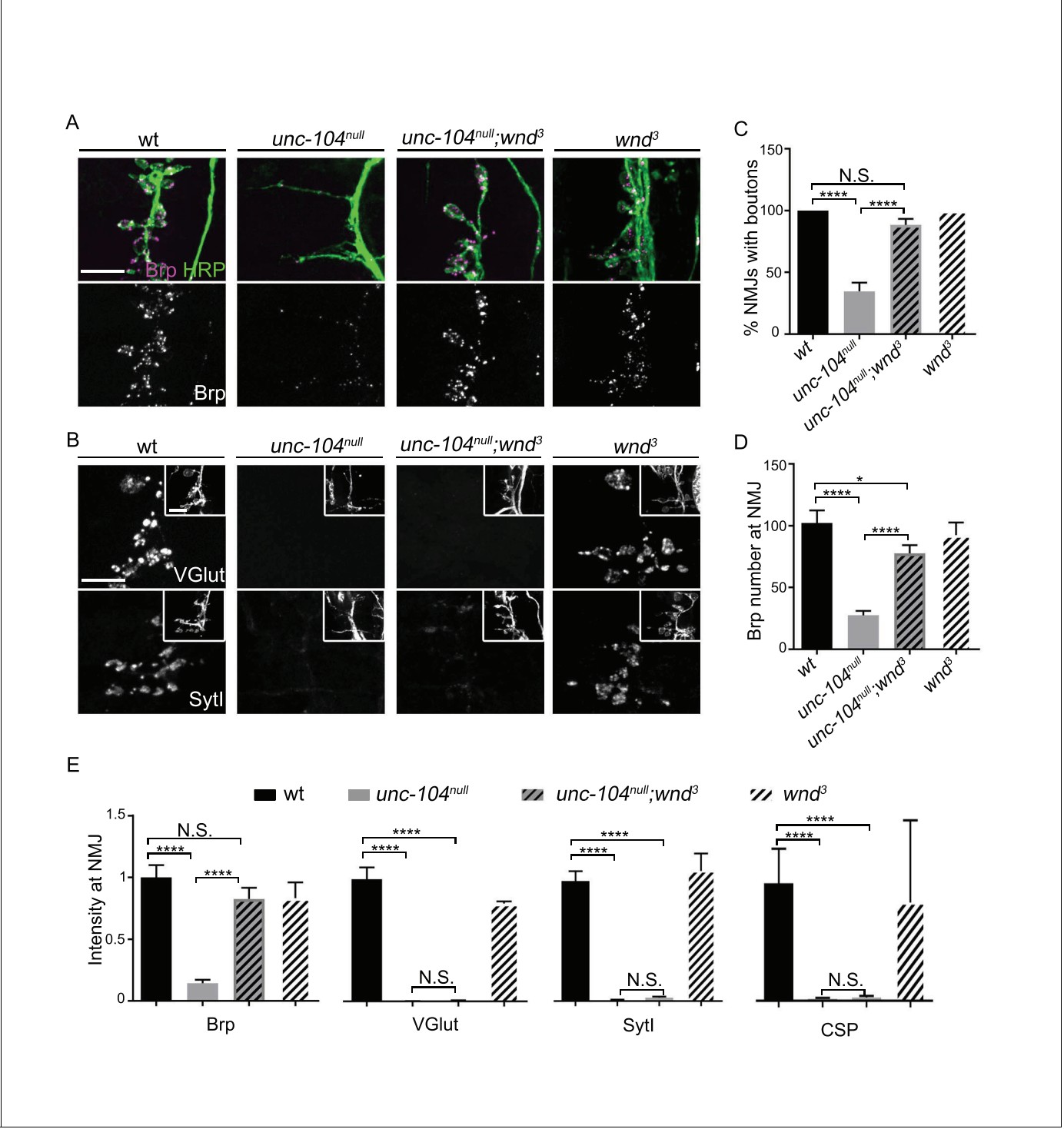

**Figure 1.** AZ transport and Synaptic bouton formation but not SVP transport defects in *unc-104-null* mutants are rescued by mutations in *wnd*. Representative images of ISNb neuromuscular junction (NMJ) terminals at muscles 6, 7, 12 and 13 at embryonic stage 17 (20–21 hr AEL). We observed similar results with multiple independently generated *unc-104-null* alleles including *unc-104^{P350}*, *unc-104^{170}* and *unc-104^{52}*. Representative images with *unc-104^{P350}* are shown here. (**A**) Images of ISNb NMJ terminals immunostained for a neuronal membrane marker to show axons and terminal boutons (HRP, green in upper panel) and the AZ component Brp (magenta in upper panel, white in lower panel). Boutons failed to form and AZs failed to localize to NMJ in *unc-104^{null}* mutants. Both defects were restored in *unc-104^{null(P350)};wnd^3* mutants. (**B**) Images of ISNb NMJ terminals immunostained for presynaptic vesicle proteins (VGlut, upper panels; SytI, lower panel). The insets show HRP staining of the nerve terminals. Note VGlut and SytI failed to localize to NMJ in both *unc-104^{null(P350)}* and *unc-104^{null(P350)}; wnd^3* mutants. (**C**) The bouton formation defect was quantified as the percentage of

*Figure 1 continued on next page*

*Figure 1 continued*

ISNb NMJ terminals that contained at least one presynaptic varicosity, identified by HRP staining (and scored while blind to genotype). This method over-estimates the actual number of boutons in unc-104[null] mutants, since any varicosity within any of the ISNb terminals (on muscles 6, 7, 12 or 13) was counted. (D) The number of AZs (identified as Brp punctae) formed at ISNb NMJ terminals (on muscles 6, 7, 12 and 13). (E) The total (sum) intensity of Brp, VGlut, SytI and CSP measured across the ISNb NMJ terminals. All data are represented as mean ±SEM; At least 9 animals and 20 ISNb NMJ terminals were examined per genotype; N.S., not significant; ****p<0.0001, *p<0.05; Tukey test for multiple comparison; Scale bar, 10 μm.
DOI: https://doi.org/10.7554/eLife.24271.003

## Suppression of *unc-104-hypomorph* defects reveals additional roles for the Wnd signaling pathway in restraining neurotransmission

In contrast with *unc-104-null* mutants, we observed that *wnd* mutations caused a striking increase in quantity of VGlut at *unc-104-hypomorphic* mutant NMJs (*Figure 2F*). The dramatic rescue of structural defects as well as SV protein localization at NMJs suggested that Wnd pathway mutations may also rescue synaptic transmission defects in *unc-104-hypomorph* mutants. *Unc-104-hypomorph* mutant NMJs have severely reduced mini frequency, and modestly reduced mEJP amplitude, EJP amplitude and quantal content (*Zhang et al., 2017*). We observed that mutations in *wnd*, *jnk* (*bsk*) or *fos* fully rescued the mEJP frequency defects (*Figure 3A and C* and *Figure 3—figure supplement 1*). mEJP amplitude, EJP amplitude and quantal content were also rescued by *wnd* mutations (*Figure 3*), while Bsk (JNK) and Fos inhibition rescued some aspects of these defects (*Figure 3—figure supplement 1* and *Figure 3—source datas 1–3*). Through distribution analysis, we further confirmed the change in mEJP amplitude and validated the coverage of mEJP events for frequency analysis (*Figure 3—figure supplement 2*). The suppression of these phenotypes by mutations in *wnd* suggests that the Wnd pathway impairs multiple aspects of synaptic structure and transmission.

The total levels of Unc-104 protein remained low in the *unc-104[hypomorph]*;*wnd* double mutants (*Figure 3—figure supplement 3*). Moreover, defects in larval motility and lethality of *unc-104-hypomorph* mutants largely remained in *unc-104[hypomorph]*;*wnd* double mutants, implying the persistence of major defects from the loss of Unc-104's function (*Figure 3—figure supplement 4*). Hence the extensive suppression of the synaptic defects at the larval NMJ is unlikely to be the result of increased stability of the residual Unc-104 protein. Instead, as indicated by the suppression of *unc-104-null* defects (*Figure 1*), the synaptic phenotypes reflect an activity of the Wnd pathway rather than a direct consequence of impaired Unc-104 driven transport.

## Wnd signaling becomes activated when Unc-104's function is lost

To test for Wnd signaling activation in *unc-104* mutants, we utilized a transcriptional reporter of JNK signaling, the *puckered* (*puc*)-lacZ enhancer trap (*Martín-Blanco et al., 1998*), which has been previously shown to report Wnd signaling activity in motoneurons (*Valakh et al., 2013*; *Xiong et al., 2010*). The basal expression of *puc*-lacZ is very low in wild type animals but was significantly increased in all *unc-104 LOF* mutant backgrounds (including *bris/null, O3.1/null,* and 2 independent RNAi lines) (*Figure 4A and B*). This increase was abolished when *wnd* was concomitantly knocked-down by RNAi (*Figure 4B*), hence reflects activation of a Wnd-mediated nuclear signaling cascade.

Additional evidence for Wnd signaling activation was revealed by the presynaptic nerve terminal morphology of *unc-104* hypomorphic mutant NMJs, which show increased numbers of synaptic branches and boutons which are smaller in size ([*Figure 4C and D*], and Kern et al). These features of presynaptic nerve terminal overgrowth have previously been described for Wnd pathway activation (*Brace et al., 2014*; *Collins et al., 2006*; *Valakh et al., 2013*; *Wu et al., 2007*). We confirmed that this overgrowth phenotype in *unc-104* mutants requires the function of Wnd, JNK and Fos in presynaptic motoneurons (*Figure 4C and D*).

Finally, we observed that *unc-104-hypomorph* mutants showed an enhanced ability to regrow axons after injury (*Figure 4E–G*). A potentially similar role in restraining axonal growth has recently been noted for *unc-104* during development in *C. elegans* (*Stavoe et al., 2016*). Since previous studies have shown that activation of Wnd/DLK signaling enhances the ability of axons to initiate regenerative axonal growth after injury (*Hammarlund et al., 2009*; *Shin et al., 2012*; *Xiong et al., 2010*; *Yan et al., 2009*), the activation of Wnd signaling in *unc-104* mutants may explain this kinesin's paradoxical function in restraining rather than promoting axonal growth.

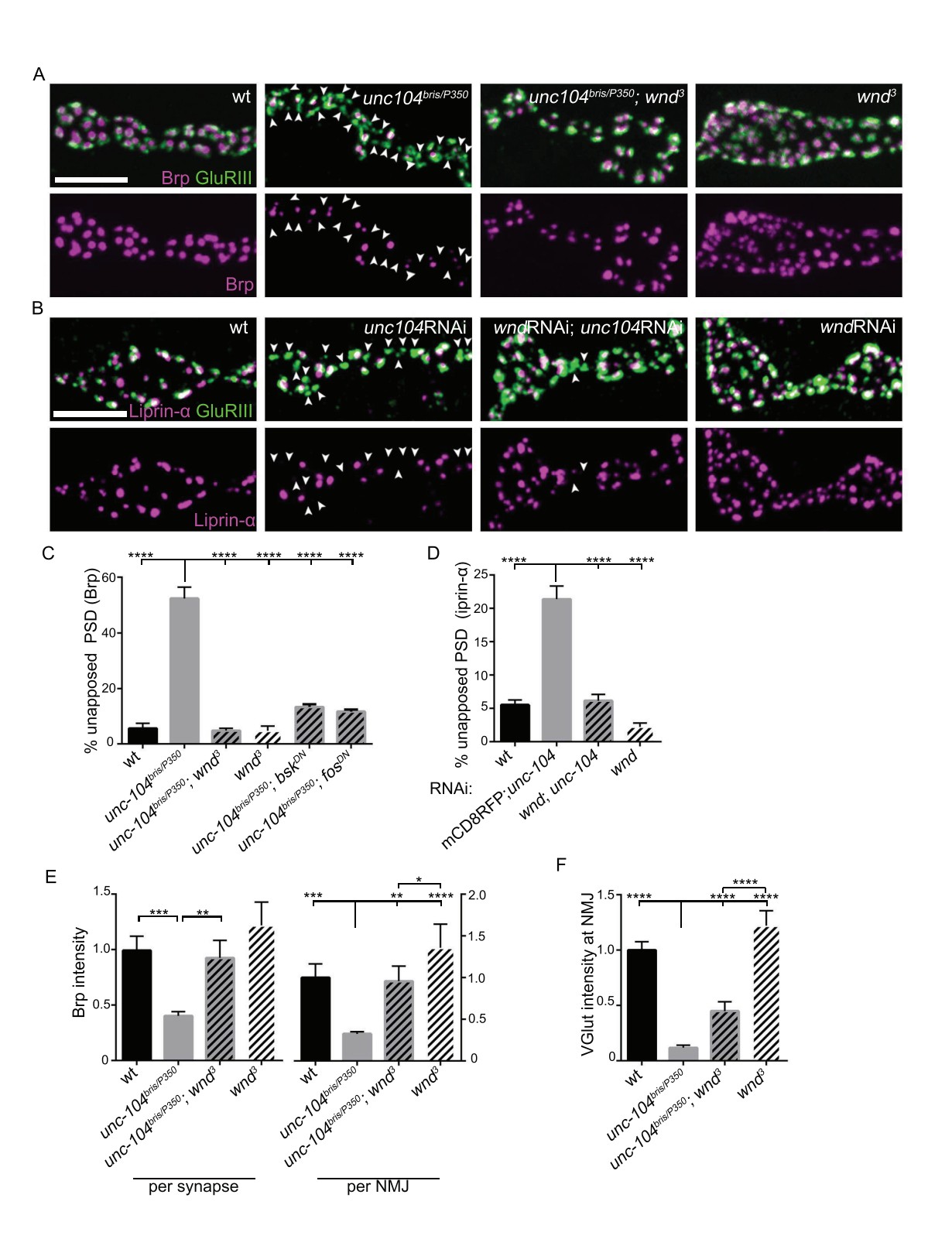

**Figure 2.** Wnd signaling pathway is required for the presynaptic defects in *unc-104-hypomorph* mutants. (A–B) Representative confocal images of third instar larval neuromuscular junctions (NMJ) at muscle 4. Postsynaptic densities (PSDs) identified by GluRIII staining (Green) that lacked apposing AZ components Brp (magenta in A), or Liprin-α (magenta in B) are highlighted by arrowheads. Note that when determining the absence of AZ components, the pixel saturation threshold was reduced to ensure detection of weak signals (this resulted in saturation of certain pixels in wt

*Figure 2 continued on next page*

*Figure 2 continued*

controls. For more details, see Experimental Procedures.). (**A**) Alignment of postsynaptic GluRIII (green) with presynaptic AZ component Brp (magenta) in *Canton-S* (wt), *unc-104*[bris/P350], *unc-104*[bris/P350];*wnd*[3] and *wnd*[3]. *Unc-104*[bris] is a partial *loss of function* mutation (*Kern et al., 2013*), while *unc-104*[P350] is a *null* allele (*Barkus et al., 2008*). *wnd*[3] is a presumptive null mutation in *wnd* (*Collins et al., 2006*). (**B**) Alignment of postsynaptic GluRIII (green) with presynaptic AZ component Liprin-α (magenta). Liprin-α-GFP was driven by rab7 promoter and UAS-*unc-104* RNAi and UAS-*wnd* RNAi were driven by neuronal *elav*-Gal4. UAS-mCD8-RFP was used as a control for UAS dosage. (**C–D**) The percentage of PSDs which lack any trace of apposing presynaptic AZ protein, based on (**C**) Brp or (**D**) Liprin-α-GFP. (**E**) Signal intensity for Brp staining at individual synapses or summed across the entire NMJ terminal (at muscle 4), normalized to that in wt (*Canton S*) animals. (**F**) The total intensity of Vglut protein measured across the entire synaptic NMJ terminal at the muscle 4, normalized to that in wt (*Canton S*). All data are represented as mean ±SEM; At least 6 animals and 12 muscle 4 NMJ terminals were examined per genotype; ****$p<0.0001$, ***$p<0.001$, **$p<0.01$, *$p<0.05$. Scale bar, 5 μm. For additional data, see *Figure 2—figure supplements 1–2*.

DOI: https://doi.org/10.7554/eLife.24271.004

The following figure supplements are available for figure 2:

**Figure supplement 1.** Wnd acts in a neuron-specific and cell-autonomous manner to mediate presynaptic defects in *unc-104-hypomorph* mutants, (related to *Figure 2*).

DOI: https://doi.org/10.7554/eLife.24271.005

**Figure supplement 2.** Wnd signaling components JNK/Bsk and Fos promotes presynaptic defects in the localization of AZs and release machinery in *unc-104-hypomorph* mutants, (related to *Figure 2*).

DOI: https://doi.org/10.7554/eLife.24271.006

Previous studies have suggested links between JNK signaling and the regulation of axonal transport (*Fu and Holzbaur, 2014*; *Verhey and Hammond, 2009*). It is therefore interesting that other mutations that disrupt axonal transport, including mutations which inhibit kinesin-1, dynein and dynactin, did not significantly affect the expression of *puc*-lacZ (*Figure 4B* and *Xiong et al., 2010*). Altogether, our results suggest a unique functional interdependence between Wnd pathway and the Unc-104 kinesin, and that the Wnd signaling pathway becomes activated when Unc-104's function is lost.

## Wnd protein is mislocalized to the cell body in *unc-104* mutants

Previous studies have established that the protein levels of Wnd and its DLK homologues in mammals and *C. elegans* are strictly regulated, and positively correlate with the activity of Wnd signaling pathway (*Feoktistov and Herman, 2016*; *Hao et al., 2016*; *Huntwork-Rodriguez et al., 2013*). A highly conserved ubiquitin ligase domain protein Hiw/Rpm-1/Phr1 restricts the levels of Wnd/DLK protein and inhibits Wnd/DLK signaling activation in wild type animals (*Babetto et al., 2013*; *Collins et al., 2006*; *Nakata et al., 2005*; *Xiong et al., 2010*). Mutations in *hiw* lead to strongly elevated levels of Wnd protein whose localization can be detected in neurites and at synapses (*Collins et al., 2006*). We therefore examined Wnd protein in *unc-104* mutants, for comparison to the previous characterized changes in *hiw* mutants.

In contrast to *hiw*, mutations in *unc-104* did not lead to a detectable increase in global levels of endogenous Wnd protein by Western Blot (*Figure 5A*), or in synaptic Wnd protein level (*Figure 5B*). In addition, *unc-104* mutants do not share the previously reported phenotype of *hiw* mutants of delayed axonal degeneration (*Xiong et al., 2012*, and data not shown). These observations suggest that the mechanism of Wnd signaling activation in *unc-104* mutants is unlikely to occur via Hiw.

We then looked further at Wnd's localization using available reagents. By immunocytochemistry (IHC) with anti-Wnd antibodies, Wnd protein remained below the detection threshold in both wild type and *unc-104* mutant animals (data not shown). Therefore, to further evaluate potential effects upon Wnd protein we utilized two different wnd-GFP fusion proteins. First, a MiMIC-*wnd*-GFP line (*Venken et al., 2011*), in which a GFP tag is inserted via an exon trap within the *wnd* genomic locus, creates another tool for detecting Wnd as expressed from its endogenous promoter. We verified that Wnd protein from this line was expressed and tagged (*Figure 5—figure supplement 1*). By IHC, this reagent revealed Wnd enrichment in neuronal cell bodies in *unc-104* mutants (*Figure 5C and E*). Second, an ectopically expressed UAS-GFP-*wnd*[KD] transgene (which was kinase dead to avoid pathway activation) has been established as a sensitive method to detect Wnd protein change (*Collins et al., 2006*; *Hao et al., 2016*; *Xiong et al., 2010*). This highly expressed transgene also revealed an increase in cell body localized Wnd in *unc-104* mutants (*Figure 5D and F*). This correlation of Wnd localization with signaling activation is interesting to note, since Wnd and its DLK

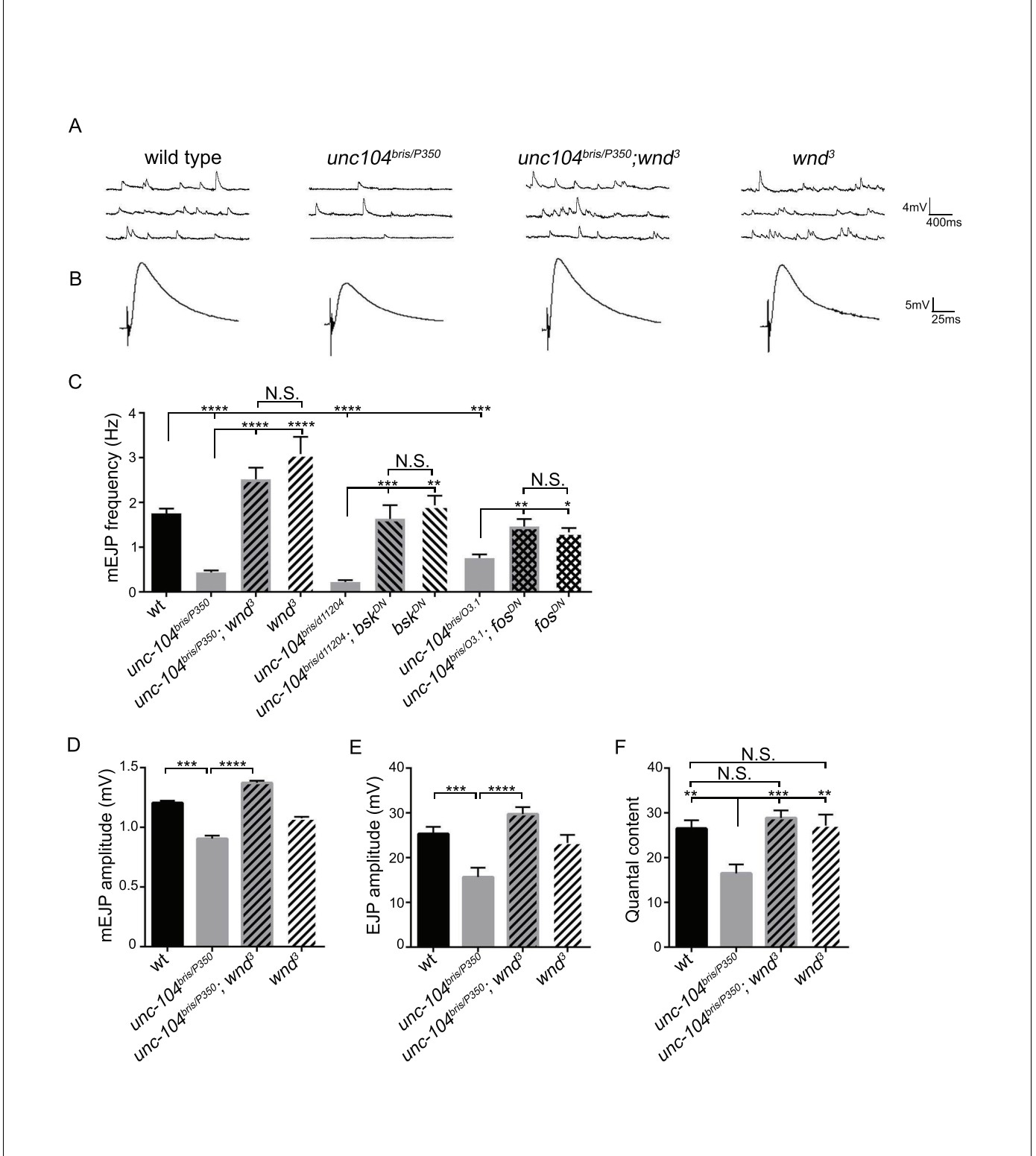

**Figure 3.** The synaptic transmission defect in *unc-104* mutants is suppressed by *wnd* mutations. (A–B) Representative electrophysiological traces of (A) miniature Excitatory Junctional Potentials (mEJP) and (B) Evoked Excitatory Junctional Potentials (EJP) recorded from muscle 6 of third instar larvae. (C) Quantification of impaired mEJP frequency for *unc-104-hypomorph* mutants and their rescue by *wnd* mutations or expression of dominant negative (DN) *bsk* or *fos* (driven by *elav*-Gal4 and *BG380*-Gal4, respectively). Representative traces for *bsk^DN^* and *fos^DN^* are shown in **Figure 3—figure supplement 1**. (D–F) Quantification of (D) mEJP amplitude, (E) EJP amplitude and (F) quantal content (corrected for nonlinear summation). See also
*Figure 3 continued on next page*

*Figure 3 continued*

*Figure 3—figure supplement 1* for additional quantification of *bsk^DN* and *fos^DN* data. Unc-104*^{03.1}* and *unc-104^{bris}* are hypomorphic alleles, while *unc-104^{P350}* and *unc-104^{d11204}* are null alleles. *wnd³* is a presumptive null mutation in *wnd* (*Collins et al., 2006*). wt animals are *Canton S*. All data are represented as mean ±SEM; N.S., not significant; ****p<0.0001, ***p<0.001, **p<0.01, *p<0.05. Tukey test for multiple comparison. For additional data, see *Figure 3—figure supplements 1–4*.

DOI: https://doi.org/10.7554/eLife.24271.007

The following source data and figure supplements are available for figure 3:

**Source data 1.** Measurements of EJP and mEJP.
DOI: https://doi.org/10.7554/eLife.24271.012
**Source data 2.** Measurements of EJP and mEJP.
DOI: https://doi.org/10.7554/eLife.24271.013
**Source data 3.** Measurements of EJP and mEJP.
DOI: https://doi.org/10.7554/eLife.24271.014
**Figure supplement 1.** Wnd signaling components JNK/Bsk and Fos promotes synaptic transmission defects in *unc-104-hypomorph* mutants, (related to *Figure 3*).
DOI: https://doi.org/10.7554/eLife.24271.008
**Figure supplement 2.** mEJP amplitude is reduced in *unc-104* mutants and restored in *unc-104; wnd* double mutants, (related to *Figure 3*).
DOI: https://doi.org/10.7554/eLife.24271.009
**Figure supplement 3.** The expression of Unc-104 is barely detectable in *unc-104-hypomorph* mutants, (related to *Figure 3*).
DOI: https://doi.org/10.7554/eLife.24271.010
**Figure supplement 4.** Defects in larval motility and lethality of *unc-104-hypomorph* mutants were only partially suppressed by *wnd* mutations, (related to *Figure 3*).
DOI: https://doi.org/10.7554/eLife.24271.011

homologue are known to physically associate with vesicles that are transported in axons (*Holland et al., 2016*; *Xiong et al., 2010*), and this transport appears to be important for its ability to mediate axon-to-cell body retrograde signaling.

Does the cell body localization of Wnd reflect a role for Unc-104 in transporting Wnd-associated vesicles? From live imaging analysis of UAS- GFP-*wnd^{KD}* vesicles we observed no impairment in transport kinetics or flux in *unc-104* mutants (*Figure 5—figure supplement 2*), and no co-localization for Wnd and Unc-104 (data not shown). Therefore we found no evidence for Wnd as a direct cargo of Unc-104.

## Wnd inhibits presynaptic structure independently of the synaptic cargo Liprin-α and Rab3

We then considered the possibility that Unc-104 regulates Wnd signaling indirectly via the transport of another cargo. Previous work has suggested that other presumed cargo of Unc-104, including Liprin-α, Rab3 and Rab3-GEF, play important early roles in AZ assembly (*Graf et al., 2009*; *Niwa et al., 2008*; *Shin et al., 2003*; *Südhof, 2012*). We therefore asked whether defects in these synaptic 'cargos' of Unc-104 led to activation of Wnd signaling. We observed that *liprin-α* mutant NMJs contain a large portion of unapposed GluRIII-labeled PSDs (*Figure 5G*), resembling the defects in *unc-104* mutants. Similar defects were reported for *rab3* and *rab3-gef* mutants (*Bae et al., 2016*; *Graf et al., 2009*). However, in contrast to *unc-104*, the *liprin-α* and *rab3* synaptic defects were not suppressed by mutations in *wnd* (*Figure 5G–I*). Furthermore, the increased Brp intensity per AZ and reduced AZ number due to *liprin-α* and *rab3* mutations was not suppressed by *wnd* mutations (*Figure 5—figure supplement 3*). In fact, the Brp intensity per AZ was slightly enhanced in *liprin-α;wnd* double mutants. These data suggest that Liprin-α and Rab3 regulate presynaptic assembly via pathways that are either independent or downstream of Wnd, and that defects in synapse assembly and structure per se do not cause activation of Wnd signaling.

## Activation of the Wnd signaling pathway in neurons is sufficient to impair presynaptic structure and synaptic transmission

If Wnd signaling activation mediates synaptic defects in *unc-104* mutants, then activation of this pathway via other means should also induce *unc-104*-like synaptic phenotypes. Indeed, over-expression of *wnd* alone in motoneurons resulted in cell-autonomous presynaptic defects that are

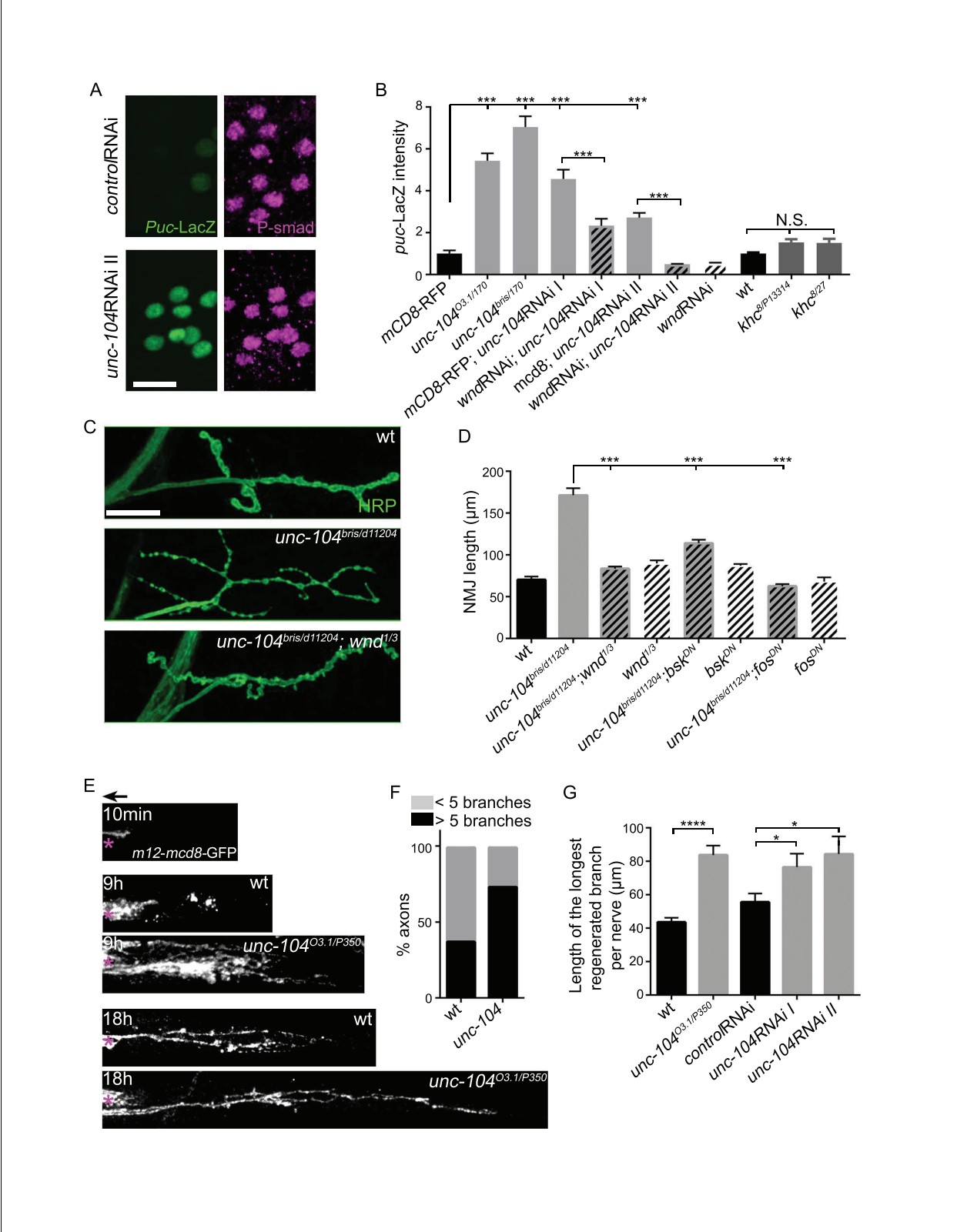

**Figure 4.** The Wnd signaling pathway is activated in *unc-104* mutants. (**A**) Images of motoneuron cell bodies near the midline of larval ventral nerve cord, immunostained for motoneuron nuclear marker phospho-SMAD (magenta) and the Wnd/JNK nuclear reporter *puc*-LacZ (green). UAS-RNAi lines (including UAS-*moody* RNAi as a control) were driven by pan-neuronal *BG380*-Gal4. At least 3 dorsal abdominal clusters of motoneurons per VNC from 6 animals per genotype were examined. (**B**) Quantification of *puc*-lacZ expression in multiple *unc-104* mutant backgrounds. *Unc-104*^O3.1^ and *unc-104*^bris^

*Figure 4 continued on next page*

*Figure 4 continued*

are hypomorphic alleles, while *unc-104[170]* is a null allele. In addition we tested two independent *unc-104 RNAi* lines (vdrc 23465, I and TRiP BL43264, II). *Unc-104* RNAi II was accompanied with Dicer2 expression to facilitate the knock-down. Co-expression of *wnd-RNAi,* compared to UAS-*mCD8*-ChRFP reduced the effect of *unc-104* knockdown. The 'control' genotype has UAS-*mCD8*-ChRFP, (which serves as a control for dosage of UAS lines). Quantification methods are described in Experimental Procedures. (**C**) Presynaptic bouton morphology and branching at the NMJ, viewed via immunostaining for HRP (membrane marker) at muscle 4. The presynaptic arbor was over-branched in *unc-104 [bris]* mutants compared to wt (*Canton S*) control animals, and this was rescued in *unc-104[bris]; wnd[1/3]* double mutants. (**D**) Quantification of presynaptic overgrowth was estimated by measuring the total NMJ length (from the most proximal to the most distal bouton of the presynaptic nerve terminal at muscle 4, labeled via anti-HRP staining). The data shown used *unc-104[bris]* and *wnd[1/3]* mutations, while UAS-*bsk[DN]* and UAS-*fos[DN]* were driven by *elav*-Gal4. wt control animals were *Canton S*. (**E**) Regenerative axonal sprouting of *m12*-Gal4, UAS-*mcd8*-GFP labeled axons 10 min, 9 hr or 18 hr after nerve crush from wt compared to *unc-104[O3.1 (hypomorph)/P350(null()]* mutant animals. Asterisk (*) indicates the injury site and arrow indicates the direction of the cell body. By 9 hr after nerve crush injury, axons in wild type animal initiate new growth via short filopodia-like branches from the proximal axonal stump. At 18 hr, a few branches are stabilized and grow either towards the distal axon or the cell body. In comparison, *unc-104* mutants showed a marked increase in new axonal branches at 9 hr, and at 18 hr these new axonal branches showed similar stabilization but extended nearly twice as far as wild type axons. (**F**) 9 hr after injury, the percentage of axons which contain more than 5 identifiable branches per axon. (**G**) The length of the longest branch per nerve at 18 hr after injury. *unc-104[O3.1/P350]* mutant and 2 independent *unc-104* RNAi lines and *control* RNAi (*moody*-RNAi) (driven by *m12*-Gal4) were examined. All data are represented as mean ±SEM; N.S., not significant; ****p<0.0001, ***p<0.001,*p<0.05, Tukey test for multiple comparison; Scale bar, 20 µm.
DOI: https://doi.org/10.7554/eLife.24271.015

comparable to *unc-104* mutants. First, many individual synapses (identified by GluRIII puncta) lacked Brp (**Figure 6A and C**), and synapses that contained Brp had reduced Brp intensity, which resulted in a global 70% reduction in Brp intensity across the entire NMJ terminal (**Figure 6B**). Second, mEJP frequency, along with other aspects of synaptic transmission (amplitude of EJP and mEJP), was significantly impaired (**Figure 6D–I**). Third, synaptic localization of SV-associated proteins, measured by VGlut intensity within NMJ terminals, was also reduced (**Figure 6B** and **Collins et al., 2006**). Taken together with the rescue of *unc-104* synaptic defects by *wnd* mutations (**Figures 1–3**), these data indicate that activation of Wnd signaling leads to strong perturbations in presynaptic structure and function.

As noted above, Wnd signaling also becomes activated when an upstream negative regulator, Hiw, is mutated (**Collins et al., 2006**; **Nakata et al., 2005**). We found that *hiw* mutants displayed strikingly similar presynaptic defects to *unc-104* mutants or overexpression of Wnd (**Figure 6—figure supplements 1** and **2**). These included 60% of synapses lacking the AZ proteins Brp, Liprin-α and Cac (**Figure 6—figure supplement 1A,B and E**), a reduction of total Brp intensity at individual synapses and across the NMJ terminal (**Figure 6—figure supplement 1C–D**), a reduction in VGlut intensity at the NMJ terminal (**Collins et al., 2006**), and reduced mEJP frequency (**Collins et al., 2006**) and **Figure 6—figure supplement 2**). All these presynaptic defects were rescued by *wnd* mutations. Meanwhile, a previously described Wnd-independent defect in quantal content remains in *hiw;wnd* double mutants (**Figure 6—figure supplement 2**, and **Collins et al., 2006**). Taken together, these observations indicate that activation of the Wnd signaling pathway in multiple scenarios—in *unc-104* mutants, in *hiw* mutants, or when Wnd is ectopically over-expressed—all lead to a shared set of characteristic defects in presynaptic structure and synaptic transmission.

## Wnd signaling restricts the expression level of presynaptic proteins in *unc-104* mutants

The above observations suggest that activation of the Wnd signaling pathway in *unc-104* mutants is the cause of many of the synaptic defects associated with Unc-104. However how Wnd signaling affects synapses, including both AZ and SV protein localization, is unknown. One of the hallmarks of the *unc-104* mutant phenotypes is the aggregation of protein components of both SVs and AZs in neuronal cell bodies (**Hall and Hedgecock, 1991**; **Pack-Chung et al., 2007**). Strikingly, *wnd* mutations enhanced this phenotype: SV and AZ components were dramatically increased in cell bodies of *unc-104; wnd* double mutants (**Figure 7A,C,E**, and **Figure 7—figure supplement 1**). We observed a similar increase when either *bsk* or *fos* was inhibited in the *unc-104* mutant background (**Figure 7— figure supplement 2**).

The exacerbated SV and AZ component accumulation in the neuronal cell bodies of *unc-104; wnd* double mutants seemed at first glance hard to reconcile with the suppression of synaptic defects in

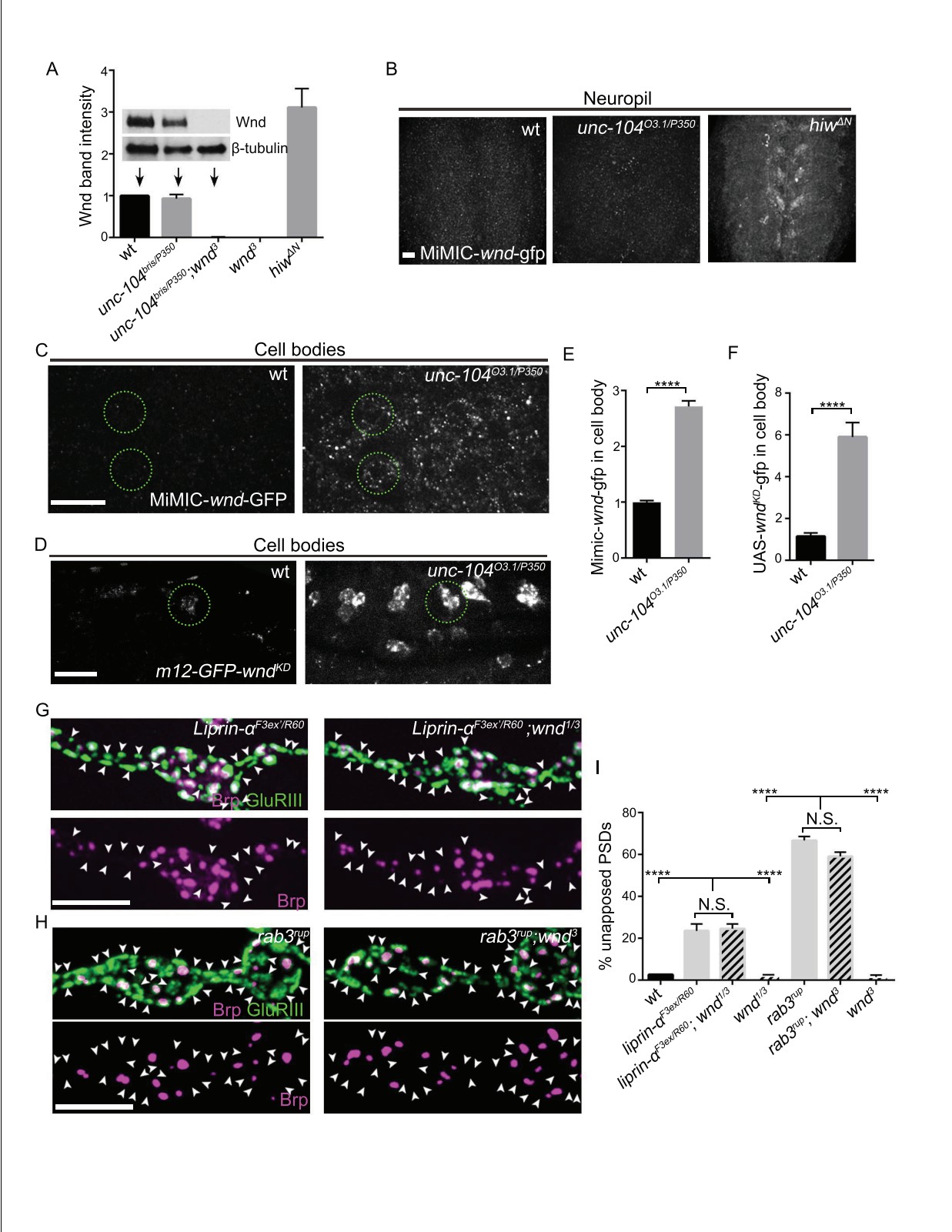

**Figure 5.** Wnd signaling is activated via an unconventional mechanism in *unc-104* mutants. (**A**) Representative Western blot of larval whole brain extracts for endogenous Wnd and β-tubulin, and quantification of Wnd levels normalized to β-tubulin band intensity (n ≥ 3). Mutants examined include *unc-104^{bris(hypomoprh)/P350(null)}*, *wnd^3* and *hiw^{ΔN}*. (**B**) Images of larval ventral nerve cords from MiMIC-*wnd*-GFP animals immunostained for GFP. Note the increased neuropil signal in *hiw* mutants, but not control or *unc-104* mutants. This also indicates that the MiMIC-*wnd*-GFP is, like endogenous Wnd

*Figure 5 continued on next page*

*Figure 5 continued*

(*Collins et al., 2006*), subject to regulation by Hiw. (**C**) Images of motoneuron cell bodies in MiMIC-*wnd*-GFP animals immunostained with antibody against GFP. Compared to (**B**), these images are collected in a higher magnification and focal plane to view the motoneuron cell bodies. Two representative cell bodies are marked by green circles. At least 6 animals were examined per genotype. (**D**) Images of SNc motoneuron cell bodies via live-imaging in larvae expressing UAS-GFP-*wnd*$^{kd}$ (which is kinase dead to avoid pathway activation) driven by *m12*-Gal4. A representative cell body is marked by a green circle. At least 6 animals were examined per genotype. (**E–F**) Quantification of GFP signal intensity from (**C–D**) in cell bodies, normalized to wt (control) animals. Note that ectopically expressed GFP-*wnd* kinase dead protein has a higher basal expression level which allows for increased sensitivity in detecting changes to Wnd protein. (**G–H**) Representative images of presynaptic Brp and postsynaptic GluRIII from (**G**) *liprin-α*$^{F3ex/R60}$ and *liprin-α*$^{F3ex/R60}$;*wnd*$^{1/3}$ and (**H**) *rab3*$^{rup}$ and *rab3*$^{rup}$;*wnd*$^3$. Unapposed GluRIII-labeled PSDs are highlighted by arrowheads. Wt is *Canton S.*. (**I**) Percentage of unapposed GluRIII-labeled PSDs from (**G**) and (**H**). All data are represented as mean ±SEM; N.S., not significant, ****p<0.0001; Tukey test for multiple comparison; Scale bar (**B–D**) 20 μm and (**G–H**) 5 μm. For additional data, see *Figure 5—figure supplements 1–3*.

DOI: https://doi.org/10.7554/eLife.24271.016

The following figure supplements are available for figure 5:

**Figure supplement 1.** MiMIC-Wnd-GFP is a fusion protein of Wnd and GFP, (related to *Figure 5*).

DOI: https://doi.org/10.7554/eLife.24271.017

**Figure supplement 2.** Wnd transport was not impaired in *unc-104* mutants, (related to *Figure 5*).

DOI: https://doi.org/10.7554/eLife.24271.018

**Figure supplement 3.** Liprin-α and Rab3 control presynaptic assembly independently of Wnd, (related to *Figure 5*).

DOI: https://doi.org/10.7554/eLife.24271.019

these animals. However, they may be explained by an action of the Wnd pathway within a feedback circuit to reduce the build-up of SV and AZ components in neuronal cell bodies. As a nuclear signaling cascade, we hypothesized that Wnd signaling may achieve this effect by down-regulating the expression levels of SV and AZ components. In this case, such reduction in global levels could account for the reduced levels at synaptic terminals in *unc-104* mutants. To test this hypothesis, we carried out quantitative immunohistochemistry to estimate the total cellular levels of Brp and VGlut in motoneurons based on respective intensities in cell body, axonal and synaptic compartments (*Figure 7F*). We found that compared with wildtype animals, the total levels of Brp and VGlut are reduced by 65% and 80% respectively in *unc-104* mutants (*Figure 7F*). The total levels of both proteins were restored in *unc-104; wnd* double mutants, with increases observed in all of the compartments (cell bodies, axons, and synapses). (*Figure 7F*). We suspect that the fact that motoneurons are only a fraction of neurons in the *Drosophila* nervous system prohibited our detection of global changes in these proteins by Western blot (data not shown). However we observed this trend in total levels of VGlut (*Figure 7—figure supplement 3*). This may be facilitated by the fact that most of the glutamatergic neurons in larval VNCs are motoneurons.

These observations, combined with the observations that Wnd activation reduces the levels of Brp and VGlut at NMJs, suggest a model in which activated Wnd signaling leads to a down-regulation of the expression levels of multiple pre-synaptic proteins in *unc-104* mutants. This response may serve to counteract their buildup in neuronal cell bodies while causing the observed defects in synapse structure and function. In support of this, *unc-104* mutations led to decreased expression of a *vglut*-DsRed transcriptional reporter in motoneurons, in a Wnd-dependent manner (*Figure 7G and H*). This, together with the involvement of transcription factor Fos in restraining VGlut and Brp buildup in cell bodies (*Figure 7—figure supplement 2*), implies that Wnd signaling may inhibit pre-synaptic protein expression at the transcriptional level.

## Wnd delays the expression of SV proteins during early stages of synapse development

We then asked whether Wnd signaling may also play a role during early synaptic development, when precise temporal coordination of presynaptic protein expression is critical for establishing synaptic contacts. Previous studies suggest that while Wnd/DLK signaling becomes activated in injured neurons, it is normally highly restrained in uninjured animals (*Collins et al., 2006*; *Xiong et al., 2010*), and, in contrast to its essential role in responses to axonal injury, roles for Wnd in developmental axonal outgrowth or synapse formation have not been previously described (*Collins et al., 2006*). We investigated this more carefully via analysis of *wnd-null* mutants during embryonic stages

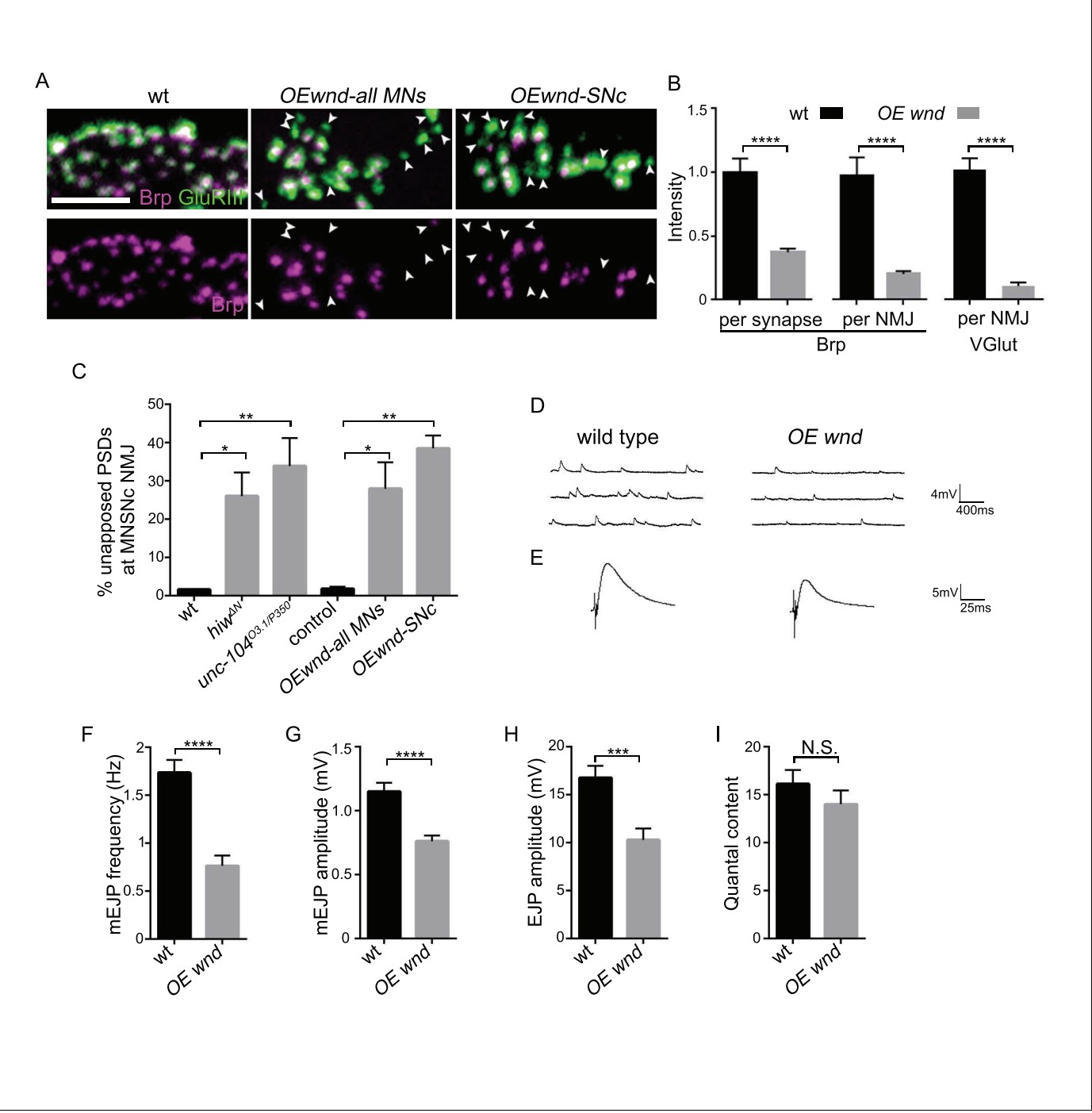

**Figure 6.** Activation of the Wnd signaling pathway in neurons is sufficient to impair presynaptic structure and synaptic transmission. (**A**) Representative images of SNc synapses at muscle 26, 27 and 29 with Brp and GluRIII staining. GluRIII-labeled PSDs that lack apposed AZs (highlighted by arrowheads) increased when Wnd was over-expressed. Similar defects were observed using a pan-motoneuron driver (*OK6*-Gal4), and in driver line specific to the SNc motoneuron (*m12*-Gal4), indicating the cell autonomy of Wnd's effect upon synapse assembly. (**B**) Quantification of Brp intensity at individual synapses or across entire NMJ terminals, and, similarly, total VGlut intensity across entire NMJ terminals (at muscle 4). Expression of UAS-*wnd* was driven with *ok6*-Gal4, and normalized to wt (*Canton S*). (**C**) The percentage of GluRIII-labeled PSDs which are unapposed by AZ component BRP. Quantification was carried out at SNc NMJ terminals in $hiw^{\Delta N}$, $unc\text{-}104^{O3.1/P350}$ and when UAS-*wnd* was over-expressed using either pan-motoneuron driver (*OK6*-Gal4) or a driver specific to SNc motoneurons (*m12*-Gal4). (**D–E**) Representative electrophysiological traces of (**D**) mEJP and (**E**) EJP on muscle 6 of third instar larvae in wt (*Canton S*) and *OE wnd* (expressed using the *ok6*-Gal4 driver). (**F–I**) Quantification of the (**F**) mEJP frequency, (**G**) mEJP amplitude, (**H**) EJP amplitude and (**I**) quantal content (corrected for nonlinear summation). All data are represented as mean ±SEM; N.S., not

*Figure 6 continued on next page*

*Figure 6 continued*

significant, ****p<0.0001, ***p<0.001, **p<0.01, *p<0.05, Tukey test for multiple comparison; Scale bar, 2 μm. For additional data, see *Figure 6—figure supplements 1–2*.

DOI: https://doi.org/10.7554/eLife.24271.020

The following figure supplements are available for figure 6:

**Figure supplement 1.** Wnd activation in *hiw* mutants inhibits presynaptic assembly, (related to *Figure 6*).

DOI: https://doi.org/10.7554/eLife.24271.021

**Figure supplement 2.** Wnd activation in *hiw null* mutants inhibits synaptic transmission, (related to *Figure 6*).

DOI: https://doi.org/10.7554/eLife.24271.022

of axonal outgrowth and synapse formation (during embryonic stages 14 through 17), and observed that NMJ synapse development progressed normally (*Figure 8—figure supplement 1*).

However, *wnd* mutants showed a premature onset of SV protein expression (*Figure 8*). In wild type embryos VGlut protein expression was undetectable until late embryonic stage 15, which coincides with the time at which motoneuron axons first reach their target muscles (*Johansen et al., 1989*). VGlut first appears in cell bodies (*Figure 8A–C*) and then becomes detectable at synapses at stage 16 and beyond (*Figure 8D*), consistent with continued expression and transport to synaptic terminals as the NMJ terminal expands. In *wnd* mutants VGlut expression levels resemble wild type at stage 16 and beyond, however its initial expression in cell bodies became apparent at an earlier time point, in early embryonic stage 15 (*Figure 8B–C*). SytI intensity was also increased in *wnd* mutants at early embryonic stage 16 (*Figure 8E–F*). These results suggest that endogenous Wnd signaling may function in pacing the onset of expression of SV proteins. This function may prevent unwanted buildup at inappropriate time points before synaptic contacts are established.

## Wnd signaling is sensitive to misregulated presynaptic proteins

Across our cumulative observations, we noticed an interesting correlation between the activity of Wnd signaling and the appearance of presynaptic proteins localized in motoneuron cell bodies. During development, the role of Wnd in restraining VGlut was most significant immediately after the onset of VGlut expression in cell bodies and before its transport to synaptic terminals (*Figure 8B–D*). The same Wnd dependent pattern was observed for SytI (*Figure 8E–F*). Similarly, in *unc-104* mutants, the highly elevated activity of Wnd signaling to restrain the expression of presynaptic components coincided with their accumulation in neuronal cell bodies (*Figure 7*).

If mislocalized or misregulated presynaptic proteins play a causal role in the induction of Wnd signaling, then over-expression of these proteins should lead to an activation of Wnd signaling independently of their impact on synaptic function. To test this, we individually overexpressed three different components pan-neuronally: Brp, SytI and VGlut. Each caused a significant induction of the Wnd responsive *puc*-lacZ reporter (*Figure 8G*). In contrast, over-expression of other proteins (Luciferase (*Figure 8G*) and membrane localized GFP (*Figure 4B*), had no effect. Over-expression of VGlut can also lead to increased synaptic transmission (*Daniels et al., 2011*), however we observed that over-expression of a non-functional VGlut transgene, VGlut$^{A470V}$, which has no effect upon synaptic physiology (*Daniels et al., 2011*), caused a similar induction of *puc*-lacZ expression (*Figure 8G*). These results, taken together with the cell autonomous nature of Wnd activation in *unc-104* mutants (*Figure 2—figure supplement 1*), suggest that Wnd signaling is sensitive to accumulations of presynaptic proteins in cell bodies.

## Discussion

### Synaptic defects in *unc-104* mutants are caused by activation of the Wnd/DLK signaling pathway

The kinesin-3 family member Unc-104/KIF1A is known to be an important mediator of synaptic development and maintenance: mutations in *unc-104* and its homologues inhibit the localization of SV and AZ precursors to nascent synapses, causing profound synaptic defects (*Barkus et al., 2008*; *Hall and Hedgecock, 1991*; *Kern et al., 2013*; *Li et al., 2016*; *Niwa et al., 2016*; *Otsuka et al., 1991*; *Pack-Chung et al., 2007*; *Yonekawa et al., 1998*; *Zhang et al., 2016*). While these synaptic

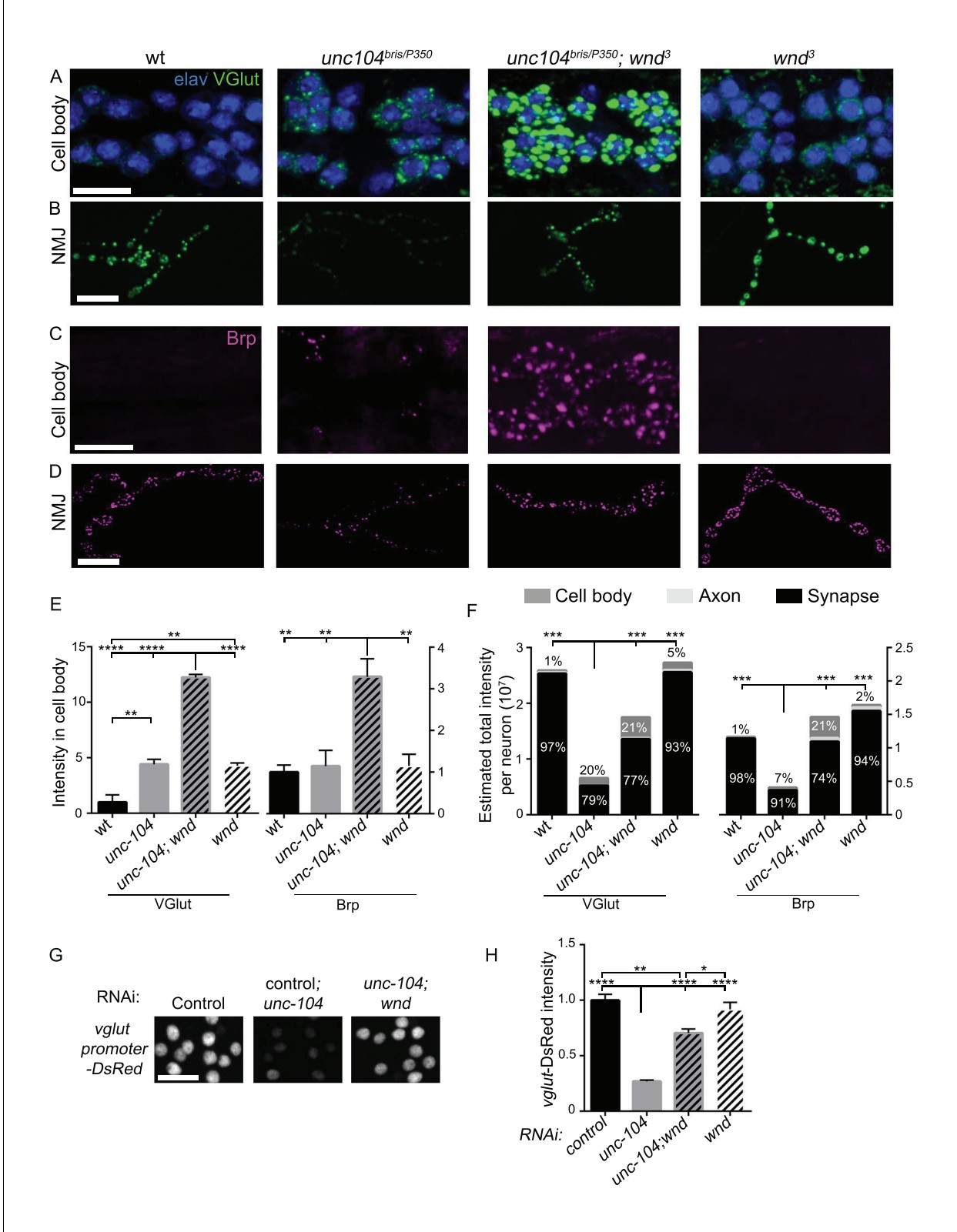

**Figure 7.** Wnd restricts the expression level of presynaptic components Brp and VGlut in *unc-104* mutants. (**A–D**) Immunostaining for VGlut (green, (**A and B**) and Brp (magenta, (**C and D**) in motoneuron cell bodies (**A and C**) and NMJ terminals (**B and D**). In (**A**) motoneuron nuclei are indicated by Elav staining (blue). Note the increased intensity of VGlut and Brp in *unc-104*^bris/P350^; *wnd*^3^ double mutants. (**E**) Quantification of VGlut and Brp intensity in cell bodies of motoneurons, corresponding to images in A and C. Note that total intensity measured at NMJ terminals is shown in *Figure 2E–F*. (**F**)

*Figure 7 continued on next page*

*Figure 7 continued*

Estimates generated for the total levels of VGlut and Brp proteins within motoneurons, accounting for the summed intensities measured in cell body, axonal, and synaptic compartments. Methods and assumptions used to calculate these estimates are described in Experimental Procedures. These proteins predominantly localize to the NMJ synaptic terminals (the proportion of the total protein localized to synapses is indicated with black shading). In *unc-104-hypomorph* mutants, a larger percentage is detected in cell bodies (medium gray shading), and the total summed intensity is reduced. In *unc-104; wnd* double mutants, the intensity increases in all of the compartments, cell body, axon (light gray shading) and synapse, compared to *unc-104* mutants alone. Wt animals are *Canton S*. (G) Images of motoneuron nuclei, immunostained for DsRed in *vglut* promoter-DsRed reporter lines. (H) Quantification of *vglut*-DsRed intensity in (G), normalized to *control*RNAi (*moody* RNAi). UAS-*RNAi* lines were driven by *OK6*-Gal4. For cell body analysis, at least 2 dorsal abdominal clusters of motoneurons per animal from at least 6 animals per genotype were examined. All data are represented as mean ±SEM; ****p<0.0001, ***p<0.001, **p<0.01, *p<0.05, Tukey test for multiple comparison; Scale bar, 20 μm. For additional data, see *Figure 7— figure supplements 1–3*.

DOI: https://doi.org/10.7554/eLife.24271.023

The following figure supplements are available for figure 7:

**Figure supplement 1.** In *unc-104-null* mutants, multiple synaptic proteins were down-regulated by Wnd, (related to *Figure 7*).

DOI: https://doi.org/10.7554/eLife.24271.024

**Figure supplement 2.** Wnd signaling components JNK/Bsk and Fos restrict the expression level of presynaptic components Brp and VGlut in cell bodies of *unc-104* mutants, (related to *Figure 7*).

DOI: https://doi.org/10.7554/eLife.24271.025

**Figure supplement 3.** Total VGlut protein levels are reduced in *unc-104-hypomorph* mutants, in a Wnd-dependent manner (related to *Figure 7*).

DOI: https://doi.org/10.7554/eLife.24271.026

defects have been considered logical outcomes of defective transport, we found that major aspects, including impaired AZ addition and development of synaptic boutons, are not mediated by a direct transport role for the Unc-104 protein. Rather, the *unc-104$^{null}$;wnd* double mutants reveal separable functions for Unc-104: (1) Transport of SV precursors to synaptic terminals is likely a direct function, since the failure to deliver adequate VGlut, Syt1 and CSP persists in *unc-104$^{null}$;wnd* double mutants, and the Unc-104 homologue KIF1A physically interacts with SV precursors (*Okada et al., 1995*). (2) Regulation of bouton formation and localization of AZs is an indirect function, since this defect can be rescued in *unc-104$^{null}$;wnd* double mutants (in the complete absence of Unc-104 function).

With the knowledge that the Wnd/DLK signaling pathway is activated in *unc-104* mutants, it is now worth considering whether it contributes to other phenotypes previously described for Unc-104 and its homologues in other species. These include impaired dendritic branching (*Kern et al., 2013*), increased microtubule dynamics (*Chen et al., 2012*), and failed neuronal remodeling (*Park et al., 2011*). It now becomes likely that these defects are mediated by Wnd/DLK, since Wnd/DLK signaling has previously been shown to influence microtubule growth (*Hirai et al., 2011; Lewcock et al., 2007*), neuronal remodeling (*Kurup et al., 2015; Marcette et al., 2014*) and dendrite growth (*Wang et al., 2013*). *Unc-104/Kif1a* mutants also show accelerated motor circuit dysfunction in aging animals (*Li et al., 2016*), impaired BDNF-stimulated synaptogenesis (*Kondo et al., 2012*) and neuronal death (*Yonekawa et al., 1998*). These phenotypes may also be facilitated by activation of DLK, which impairs synaptic development and function (this study and *Nakata et al., 2005*), and has also been shown to mediate neuronal death in some contexts (*Chen et al., 2008; Pozniak et al., 2013; Welsbie et al., 2013; Welsbie et al., 2017*). Human mutations in *KIF1A* have been associated with hereditary spastic paraplegia (SPG30) (*Fink, 2013*), and hereditary sensory and autonomic neuropathy type IIC (HSN2C) (*Rivière et al., 2011*). The possibility that DLK activation mediates deleterious aspects of these disease pathologies becomes an interesting future question.

## The Wnd/DLK pathway is sensitive to defects in Unc-104-mediated transport

How does the Wnd pathway become activated in *unc-104* mutants? The mechanism(s) that lead to activation of Wnd and its DLK homologues are of general interest for their roles in axonal regeneration as well as degeneration and neuronal death. In addition to axonal injury (*Watkins et al., 2013; Welsbie et al., 2013; Xiong et al., 2010*), disruption of microtubule and/or actin/cortical cytoskeleton can lead to activation of DLK (*Valakh et al., 2013, 2015*). Moreover, many studies have noted a role for DLK signaling in mediating structural changes in neurons downstream of manipulations that disrupt cytoskeleton (*Bounoutas et al., 2011; Marcette et al., 2014; Massaro et al., 2009*). Since

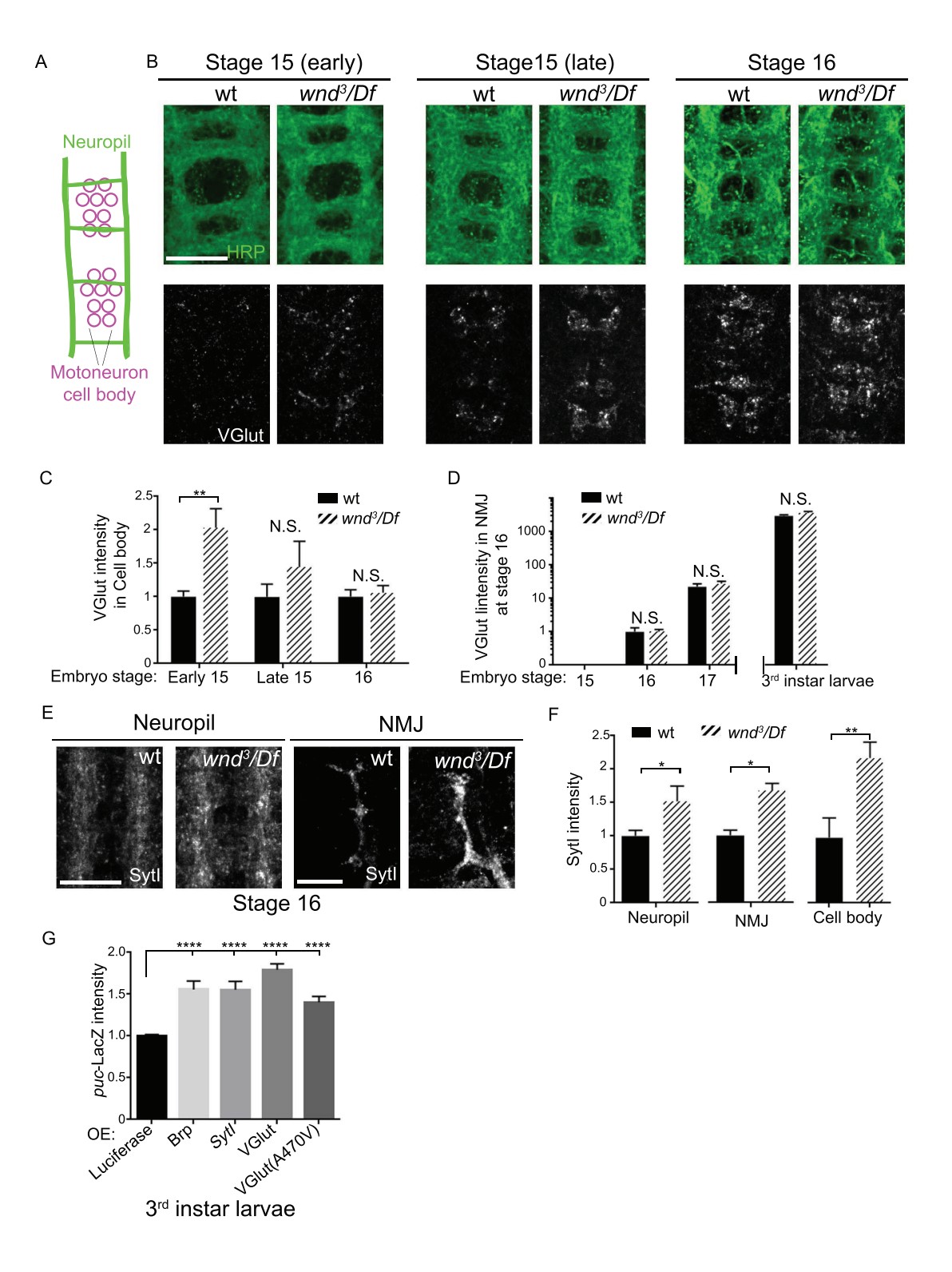

**Figure 8.** Wnd delays the expression of SV proteins during early stages of synapse development. (**A**) Schematic cartoon of the embryonic nerve cord showing the neuropil (the location of neurites and developing synapses in the CNS) and motoneuron cell bodies in 2 segments at late embryonic stages (15 to 16). (**B**) Representative images of VGlut (white, bottom row) expression in motoneuron cell bodies of wt (*w118*) and *wnd-null* mutants (*wnd³/Df*) at embryonic stage early 15, late 15 and 16. Analogous segments are identified by neuropil HRP staining (top row). (**C–D**) Quantification of
*Figure 8 continued on next page*

*Figure 8 continued*

VGlut intensity in (C) motoneuron cell bodies and (D) NMJ presynaptic terminals for wt and *wnd-null* mutants at different embryonic stages and in third instar larvae. VGlut expression first appears in cell bodies at embryonic stage 15 (C), corresponding with the onset of NMJ synaptogenesis, but does not appear at NMJ terminals until stage 16 (D). As the NMJ matures and expands throughout development, VGlut intensity, which is predominantly localized to NMJ terminals, continues to increase (note the logarithmic scale). Quantification in both (C) and (D) is normalized to intensity for wt (*w118*) animals at stage 16. (E) Representative images of SytI immunostaining in CNS neuropil and NMJs. SytI intensity is elevated in *wnd$^3$/Df* mutants at embryonic stage 16. (F) Quantification of SytI intensity from (E), normalized to intensity for wt (*w118*) animals. (G) Quantification of *puc*-lacZ expression in third instar larvae that ectopically express luciferase or presynaptic components (Brp, SytI, VGlut and VGlut$^{A470V}$) via the Gal4/UAS system. Pan neuronal Gal4 (*BG380*) was used to drive expression of UAS-lines. All data were represented as mean ±SEM; ****p<0.0001, **p<0.01, *p<0.05, Tukey test for multiple comparison; Scale bar, 20 µm. For additional data, see *Figure 8—figure supplement 1*.

DOI: https://doi.org/10.7554/eLife.24271.027

The following figure supplement is available for figure 8:

**Figure supplement 1.** Brp and VGlut localization to NMJ terminals through early stages of synaptic development are not altered in *wnd null* mutants, (related to *Figure 8*).

DOI: https://doi.org/10.7554/eLife.24271.028

the cytoskeleton is a closely functioning partner of all motor proteins, and is also implicitly affected by axonal injury, it is possible that these manipulations share a similar underlying mechanism with that of *unc-104* mutations. While disruption of cytoskeleton should impair transport by many motor proteins, mutations that impair kinesin-1 and dynein do not lead to activation of Wnd (*Figure 4* and *Xiong et al., 2010*). This specificity suggests that disruption of Unc-104 mediated transport, potentially via mislocalization of Unc-104's cargo, mediates Wnd/DLK's activation after cytoskeletal disruption and potentially after axonal injury.

This line of reasoning leads to further consideration of Unc-104's cargo. Our live imaging data do not support a simple model that Wnd is a cargo of Unc-104 (*Figure 7—figure supplement 2*). Known cargo of Unc-104 are important for the assembly and function of synapses (*Goldstein et al., 2008*), so does Wnd activation occur in response to an impairment in synaptic assembly or function? We think this is unlikely, since mutations in *rab3-gef*, *liprin-α* and *vglut*, which impair presynaptic assembly and function, do not cause activation of Wnd (data not shown).

Instead, we note an intriguing correlation between the localization and abundance of presynaptic proteins with Wnd's activation: Wnd signaling becomes activated in *unc-104* mutants, which accumulate presynaptic proteins in the neuronal cell body. We noticed a similar role for endogenous Wnd in wild type neurons during the onset of embryonic NMJ development (*Figure 8*). These stages correspond to the onset of synaptic protein expression, before substantial transport to synaptic terminals, hence represent a time in which levels are high in the cell body. Consistent with the idea that Wnd signaling is sensitive to mislocalized presynaptic proteins, ectopic overexpression of several different presynaptic proteins caused an elevation in Wnd signaling in uninjured neurons (*Figure 8G*). These observations suggest a model that accumulations of presynaptic proteins, as a feature of aberrant cargo transport, are 'sensed' by Wnd signaling (*Figure 9*).

## Wnd/DLK signaling restrains the expression of presynaptic proteins: mechanism and relevance

While previous studies in *C. elegans* (*Nakata et al., 2005*; *Yan et al., 2009*) have suggested that DLK activation may impair synaptic development (altering the size and spacing of active zones), the regulation of total levels of presynaptic proteins and the relationship with Unc-104 provides a new view into Wnd/DLK's function and mechanism. In addition to VGlut, SytI, CSP-1, Brp, Liprin-α and Cac, we suspect that Wnd signaling restrains the expression of a cohort of presynaptic proteins. Regulation of multiple targets required for synapse development and maintenance can explain the severe defects in *unc-104-hypomorph* mutants, and the dramatic suppression by disruption of the Wnd pathway (*Figures 2* and *3*). In support of this idea, presynaptic defects in *unc-104-hypomorph* mutants can be partially rescued by overexpressing Brp (*Kern et al., 2013*) or Rab3 (*Zhang et al., 2016*).

It is interesting to consider that the targets of Wnd regulation are also abundant presynaptic proteins, and are thought to be major cargo for axonal transport (*Figure 9*). Down-regulation of these proteins in response to defects in their transport or after axonal damage may comprise a stress

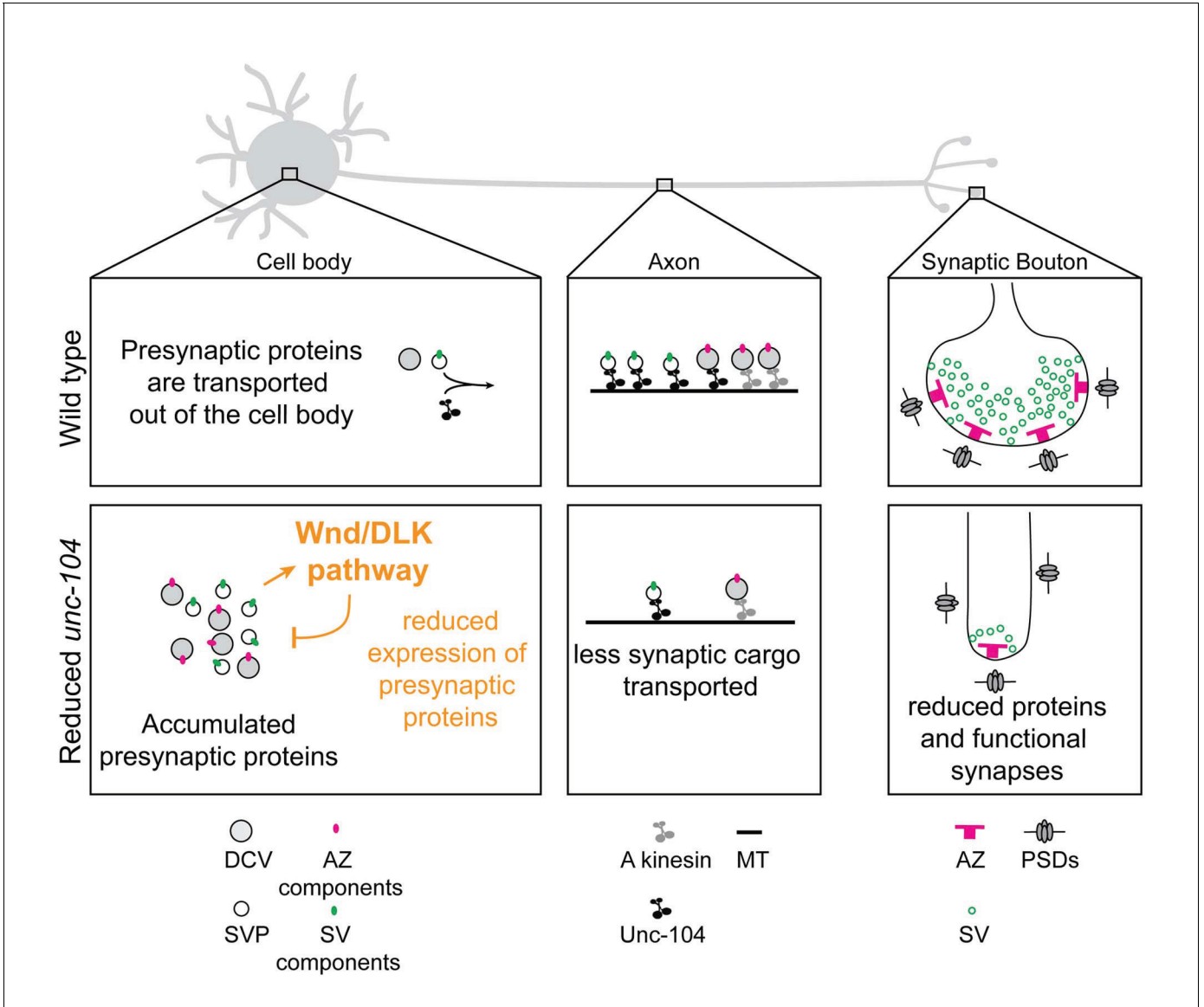

**Figure 9.** Model: Wnd/DLK signaling promotes synaptic defects by restraining total levels of presynaptic proteins when Unc-104's function is reduced. The formation and maintenance of synapses require the synthesis of presynaptic components (including AZ components (magenta dots) and SV components (green dots), which ultimately assemble into mature AZs (magenta 'T' bars) and SVs (green circles) at synapses. SV components are transported in the form of SV precursors (SVPs), while AZ components are thought to be transported via association with dense core vesicles (DCVs) (*Shapira et al., 2003*). While much remains to be learned about the mechanism by which individual synaptic components are transported, we can infer from this study that the Unc-104 kinesin (black) plays an essential role in the transport of SV components but not AZ components (which may be carried by an additional kinesin, indicated in gray). When Unc-104's function is impaired/reduced, presynaptic components accumulate in cell bodies. However Wnd signaling activation reduces the accumulations by reducing total expression levels of presynaptic components. This reduction eventually manifests as a reduction in synapse number and mismatched pre- and postsynaptic structures.

DOI: https://doi.org/10.7554/eLife.24271.029

response mechanism to prevent unwanted buildup or wasted cellular resources. This role may allow neurons adapt to stresses that impair axonal transport by counteracting cargo buildup. But the resulting reduction in synaptic protein levels can also be maladaptive, leading over time to synapse failure and/or loss. This feature may contribute to synaptic pathologies associated with defects in axonal transport.

How does Wnd signaling regulate presynaptic proteins? The regulation of the *vglut*-promoter-DsRed reporter and requirement for the Fos transcription factor suggests the involvement

transcriptional regulation (*Figure 7G–H*), and we also noticed mildly increased levels of Brp, VGlut and Cac transcripts in *unc-104;wnd* double mutants (data not shown). However we are limited in our ability to detect total changes in mRNA and protein levels from whole nerve cord preparations by the fact that the Wnd signaling pathway may not be acting in all cell types. It is also possible that additional post-transcriptional mechanisms, such as regulation of protein stability or translation, or bulk turnover via autophagy, factor into the regulation of presynaptic proteins by Wnd. A recent study has suggested that DLK may activate the integrated stress response pathway in mammalian neurons (*Larhammar et al., 2017*). On the other hand, previous studies have separately linked Unc-104, Wnd signaling and AP-1 (Fos) to autophagy (*Guo et al., 2012*; *Shen and Ganetzky, 2009*; *Stavoe et al., 2016*). It will be interesting to determine whether translation and/or autophagy contribute to the regulation of presynaptic proteins by Wnd signaling.

Many previous studies have reported links between JNK signaling and kinesin-driven transport (*Verhey and Hammond, 2009*), with some observations suggesting that JNK signaling may directly regulate the function of kinesin-1, modulating its cargo binding, affinity for microtubules and its processivity (*Fu and Holzbaur, 2013*; *Horiuchi et al., 2007*; *Morfini et al., 2006*; *Stagi et al., 2006*; *Sun et al., 2011*). Our finding of a separate role for JNK signaling in regulating the abundance of transported cargo adds a new layer of complexity to interpreting phenotypes of axonal transport defects. A commonly described defect is the presence of accumulations of cargo within axons, referred to as 'traffic jams'. These defects have been noted for many different mutations, including kinesin-1 and dynein subunits (*Gindhart et al., 1998*; *Hurd and Saxton, 1996*; *Martin et al., 1999*), and also in mutants for *wnd* and other members of JNK signaling pathways (*Bowman et al., 2000*; *Horiuchi et al., 2005*, *2007*). Does a failure to regulate excess protein cargo contribute to the presence of the jams? Intriguingly, *unc-104* mutations do not cause this type of 'traffic jams' in axons, but instead leads to accumulations of synaptic proteins in cell bodies, which correlates with the activation of Wnd signaling.

Finally, it is interesting to compare Wnd's role in tuning levels of presynaptic proteins with previously identified roles for DLK in promoting cell death (*Chen et al., 2008*; *Huntwork-Rodriguez et al., 2013*; *Pozniak et al., 2013*; *Watkins et al., 2013*; *Welsbie et al., 2013*). While *Kif1a* mutant mice show early signs of neuronal death and degeneration, which may potentially be mediated by activation of DLK, *unc-104* mutants in *C. elegans* and *Drosophila* lack hallmarks of cell death and synaptic degeneration (*Hall and Hedgecock, 1991*; *Kern et al., 2013*; *Pack-Chung et al., 2007*). In analogy with other stress response pathways, regulation of presynaptic proteins may comprise a first order response that may facilitate adaptation to stress, while cell death could be the consequence when compensatory mechanisms fail. Inhibition of synaptic proteins is, alone, a pathology that becomes relevant for long term maintenance of synapses and their function over time, since synthesis and transport of new synaptic proteins likely needs to occur throughout the long lifespan of a neuron. Interestingly, previous studies have linked activation of JNK signaling to synapse loss in aged animals (*Ma et al., 2014*; *Sclip et al., 2014*; *Voelzmann et al., 2016*). Exciting future work lies ahead to further understand DLK's activation and its consequences in different models of neuronal injury, disease, and aging.

## Note added at proof

Since acceptance of this work, a new publication has shown that DLK signaling contributes to pathology in multiple mouse models of neurodegenerative diseases (*Le Pichon et al., 2017*). This study provides further suggestive evidence that DLK signaling activation can contribute to synapse loss in disease conditions.

## Materials and methods

### *Drosophila* stocks

The following strains were used in this study: *Canton-S*, *hiw*[AN] (*Wu et al., 2005*), *wnd*[1], *wnd*[3], *wnd*[dfED228](*Collins et al., 2006*), UAS-*wnd*[kinase dead]-GFP(*Xiong et al., 2010*), MiMIC-*wnd*-GFP (*Venken et al., 2011*), *unc-104*[O3.1], *unc-104*[P350] (*Barkus et al., 2008*), *unc-104*[bris] (*Medina et al., 2006*), *unc-104*[d11204] (*Thibault et al., 2004*), *unc-104*[52] (*Pack-Chung et al., 2007*); UAS-*cacophony*-GFP(*Kawasaki et al., 2004*), OK6-Gal4 (*Aberle et al., 2002*), OK319-Gal4, OK371-Gal4(*Mahr and*

*Aberle, 2006*), *m12*-Gal4(*Ritzenthaler et al., 2000*), *BG380*-Gal4(*Budnik et al., 1996*),*elav*-Gal4[C155](*Lin and Goodman, 1994*), UAS-*fos*[DN] (*Eresh et al., 1997*), UAS-*bsk*[DN] (*Weber et al., 2000*),*khc*[8], *khc*[27](*Brendza et al., 1999*), *khc*[k13314](*Spradling et al., 1999*), *Liprin-α*[F3ex15], *Liprin-α*[R60](*Kaufmann et al., 2002*), UAS-VGlut-GFP, UAS-VGlut[A470V]-GFP (*Grygoruk et al., 2010*), UAS-Brp-GFP (Bloomington (BL) 36291 and 36292), UAS-SytI-GFP (BL6925 and 6926), Rab7-*liprin-α*-GFP (*Zhang et al., 2016*), *Rab3*[rup](*Graf et al., 2009*), UAS-YFP-*Rab3*, UAS-YFP-*Rab3*[Q80L], UAS-YFP-*Rab3*[T35N](*Zhang et al., 2007*), uas-mcd8-ChRFP (Schnorrer, 2009.5.11), *puc*-lacZ[E69](*Martín-Blanco et al., 1998*), *vglut* promoter-DsRed (gifts from Daniels and Diantonio), RNAi lines: *moody* RNAi (control), *Octβ2R* RNAi (vdrc 104524, control), *unc-104* RNAi (vdrc 23465, I and TRiP BL43264, II), *wnd* RNAi (vdrc 103410 and vdrc 26910), *Rab3* RNAi (TRiP BL31691 and BL34655), UAS-*Dcr2* was a gift from Stephan Thor (Linköping Université, Linköping Sweden). Flies were raised at 25°C or 29°C (as indicated for certain RNAi knock-down) on standard Semidefined yeast-glucose media (*Backhaus et al., 1984*).

To generate *vglut*-DsRed reporter flies, genomic sequence spanning 5.3 upstream of the ATG start codon for *vglut* (CG9887) was cloned into a plasmid derived from pCaSpeR-AUG-bGal (*Thummel et al., 1988*), in which lacZ was replaced with DsRed.T4-NLS (*Barolo et al., 2004*) coding sequence, such that the expressed DsRed would concentrate in the nucleus.

## Immunocytochemistry

Third-instar larvae were dissected in ice-cold PBS, then fixed in 4% formaldehyde (FA) in PBS/HL3 solution for 3 min for Cac-GFP, 10 min for Brp and GluRIII/GluRIIC staining or 20 min for other antibody staining, followed by blocking in PBS with 0.1% Triton (PBT) containing 5% Normal Goat Serum (NGS) block for 30 min. Control and experimental animals were always dissected, fixed and stained in the same condition and imaged in parallel using identical confocal settings.

Embryos were dissected, fixed and stained as described in (*Featherstone et al., 2009*; *Lee et al., 2009*). In brief, embryos were collected for 30–60 min on Molasses plates and kept in 18°C (for stage 14 to 16) or 25°C (for stage 17) overnight. Early-stage embryos (14-16) were dechorionated, sorted (based on GFP), staged (based on gut morphology [*Hartenstein, 1993*]) and dissected (tungsten needles) on negatively charged slides. Stage 17 embryos (20–21 hr AEL) were dissected with Vet glue (Vetbond) in PBS (PH = 7.3) on Sylgard-coated coverslips. Bouin's fixation for 5 min was used for all antibodies staining but Synapsin staining (4% PFA for 25 min). The examined *unc-104-null* alleles include *P350/P350* and *52/52*.

Primary antibody and secondary antibody incubations were conducted in PBT containing 5% NGS at 4°C overnight and at room temperature for 2 hr, respectively, with three 10 min washes in PBT after each antibody incubation. The following primary antibodies and dilutions were used: ms anti-Brp (DSHB Cat# nc82 Lot# RRID:AB_2314866), 1:200; ms anti-Synapsin (DSHB Cat# 3C11 (anti SYN-ORF1) Lot# RRID:AB_528479), 1:50; ms anti-CSP-1 (DSHB Cat# DCSP-1 (ab49) Lot# RRID:AB_2307340), 1:100; ms anti-DLG(DSHB Cat# 4F3 anti-discs large Lot# RRID:AB_528203), 1:1000; Rb anti-GluRIII (gift from Diantonio lab), 1:2500; Rb anti-SytI (gifts from Noreen Reist, *Mackler et al., 2002*), 1:400; Rb anti-Unc-104(Pack-Chung E; Nat Neurosci. 2007 Cat# unc-104 Lot# RRID:AB_2569094), 1:500; ms anti-lac-Z (DSHB Cat# 40-1a Lot# RRID:AB_528100), 1:100; Rb anti-Phospho-Smad1/5 (Cell signaling),1:100; Rb anti-DsRed (Clontech Laboratories, Inc. Cat# 632496 Lot# RRID:AB_10013483Clontech), 1:1000; Rat anti-elav (DSHB Cat# Rat-Elav-7E8A10 anti-elav Lot# RRID:AB_528218), 1:50; A488 rabbit anti-GFP (Molecular Probes Cat# A-21311 also A21311 Lot# RRID:AB_221477), 1:1000 and Alexa488/cy3/Alexa647 conjugated Goat anti-HRP (Jackson ImmunoResearch), 1:300. Rabbit anti-VGlut (gift from Diantonio lab), 1:10000, staining was carried as described in *Daniels et al. (2008)*. For secondary antibodies we used Cy3- or A488-conjugated goat anti-rabbit or anti-mouse 1:1000 (Invitrogen).

## Imaging and analysis

Confocal images were collected as described in (*Füger et al., 2012*; *Xiong et al., 2010*). Similar settings were used to collect all compared genotypes and conditions.

The identification and quantification of the % unapposed PSD was based on manual counts of the total number of individual GluRIII-labeled puncta (on either muscle 4 or 26/27/29, where indicated), scored for the presence or absence of an apposing AZ component (Brp or Liprin-α-GFP). To affirm

that AZ components were indeed completely absent, confocal settings and brightness levels were optimized for the weakest signals in *unc-104* mutants. Since the same settings were used for all genotypes some pixels for AZ components were necessarily over-exposed in the wt controls. For measurements of intensity levels, using Volocity software, only raw images acquired together using the same confocal settings were compared. To measure VGlut and Brp levels in axons and NMJs, we used staining for HRP (which labels neuronal membrane) to define the region of interest. For cell bodies, we selected the signal above a specified threshold, (To specify the threshold, we checked the background signal and examined the fluorescence signal distribution from multiple images. The threshold was chosen to be as at least 3 fold higher than the background and 1 SD higher than the center of the distribution. The same threshold was applied to all compared images). To estimate the total VGlut or Brp level within a single motoneuron (*Figure 6*), we summed measurements of: (a) total intensity for individual cell bodies (located in the dorsal midline of the ventral nerve cord), (b) total intensity for individual NMJ nerve terminals at muscle 4, and (c) estimated total intensity within a motoneuron axon, calculated from mean intensity in axonal segments, based on the assumption of 32 motoneuron axons per nerve and an average axon length of 1 mm.

When imaging nerve cord, we used 0.8 µm step size for the z-stack and focused on the posterior and central nerve cord, corresponding to A4-A8. When imaging axons or the NMJ, we used 0.4 µm step size for z-stacks. Axonal segments were imaged 900 µm away from the nerve cord. NMJ images were collected at segment A3 for muscle 4 or (when using the m12Gal4 driver) for muscle 26, 27 and 29, which are innervated by the SNc neuron. *puc*-lacZ level was measured within the nucleus region for motoneurons, selected by P-smad staining in the dorsal regions of A4-A8 in the ventral nerve cord.

For live imaging analysis of GFP-wnd-KD transport, third instar larvae were dissected in the center of a circle reinforcement label (Avery) in HL3 solution (*Stewart et al., 1994*) with 0.45 mM calcium. Larvae were pinned at the head, tail and 2 lower corners, the pins were then pushed into sylgard so that the coverslip could lie directly on top of the reinforcement label (and the larva), and excess HL3 solution was removed before imaging on an inverted microscope. Images were collected at 0.3 Hz for 5 min at 40x magnification in segmental nerves at a location 900 µm distal to the nerve cord. The images were then processed in imageJ with a kymograph plugin (Jens Rietdorf and Arne Seitz) and further analyzed in MATLAB with a program written to determine vesicle segmental speed and duration (described in *Ghannad-Rezaie et al., 2012*).

## Western blot

25–30 3rd brains were collected in PBS and homogenized for each sample. The following antibodies and dilutions were used: rb anti-Wnd 4–3 (*Collins et al., 2006*) 1:700; rb anti-VGlut (*Daniels et al., 2008*, 1:10000; ms anti-Brp (NC82), 1:100; ms anti-β-tubulin (1E7, DSHB), 1:1000; and rb anti-unc-104 (Gift from Tom Schwarz lab), 1:500. The blots were probed with HRP conjugated secondary antibodies: Gt at-ms and Gt at-rb at the dilution of (1:5000) and imaged with either film or an Odyssey CLx imager (LI-COR).

## Axonal regeneration

The nerve crush assay was carried out as described (*Xiong et al., 2010*), and animals were fixed either 9 or 18 hr after the injury. Axonal regeneration (sprouting) was quantified by measuring the number of injured axons that contained more than 5 branches at 9 hr, and the length of the longest branch at 18 hr.

## Electrophysiology

Third instar larvae were dissected within 3 min in HL3 solution containing 0.65 mM Calcium at 22°C. Muscle 6 at segment A3 was located by the use of an OLYMPUS BX51WI scope with a 10x water objective and then recorded intracellularly with an electrode made of thick wall glass (1.2 mm x 0.69 mm) pulled by SUTTER PULLER P-97. Amplifier GeneClamp 500B and digitizer Digidata 1440A were used. The recording was only used if the resting potential was negative to −60 mV and muscle resistance was >5 mΩ. A GRASS S48 STIMULATOR was used to obtain a large range of stimulation voltage range (1–70V). We noticed that *hiw* mutants and *unc-104* mutants required a higher stimulus to recruit the 2nd axon that intervenes Muscle 6 (10–40V were required in *hiw* and *unc-104* mutants, as

opposed to 2–8V in wild type). To ensure that we could always recruit both axons, for each muscle we tested a range of stimulation voltage (1–70V) to find the threshold which triggered the largest response within the testing range. A stimulus slightly larger than this threshold was then used at a frequency of 0.2 Hz and duration of 1 ms for EJP measurements. Axon Laboratory software was used for acquisition and the Mini Analysis program (Synaptosoft Inc) was used for analysis of mEJP frequency and amplitude. Parameters for Mini Analysis were set as: 0.2 (threshold), 1 (area threshold), 30,000 (period to search a local maximum), 40,000 (period to search a decay time), 40,000 (time before a peak for baseline), 20,000 (period to average a baseline), 0.6 (fraction of peak to find a decay time) and detect complex peak.

Quantal content and EJP amplitudes were corrected for non-linear summation using the revised Martin correction factor as described in (*Kim et al., 2009*; *Morgan and Curran, 1991*). In brief, to accurately estimate the potential change if units sum linearly, the following equation was applied when EJP amplitude was larger than 15% of $E^{resting\ potential}$: corrected EJP = EJP/(1 f(EJP/$E^{driving\ force}$)), where $E^{driving\ force}=E^{resting\ potential}-E^{reversal\ potential}$ and f = the membrane capacitance factor ($\Delta t/\tau$). At the Drosophila NMJ, $E^{reversal\ potential}$ is estimated around 0 mV, $E^{resting\ potential}$ from our recordings is around −70 mV and f = 0.55 (*Kim et al., 2009*; *Morgan and Curran, 1991*).

## Data analysis

The minimum sample size for animal lethality and motility was determined by power analysis using: sample size = $2*SD^2(Z_{\alpha/2}+Z_\beta)^2/d^2$, where SD = standard deviation, d = effect size and $Z_{\alpha/2}$ or $Z_\beta$ are constants when $\alpha$ = 0.5 and $\beta$ = 80% (*Jung, 2010*). For other assays (including synapse morphology, *puc*-lacZ expression and axonal sprouting), the sample size and biological replicate number was determined for comparison with previous studies (*Xiong et al., 2010*; *Xiong et al., 2012*; *Collins et al., 2006*; *Kern et al., 2013*; *Ghannad-Rezaie et al., 2012*). For axonal regeneration (sprouting), a total of 30 axons were measured from 8 individual animals. For comparison of intensities via confocal imaging and for structural analysis of synapse number and bouton number, at least 12 clusters of motoneurons within dorsal abdominal segments 2–5 (which contain 10 cell bodies per cluster), taken from at least 6 animals (2–3 clusters per animal) or 12 NMJs from at least 6 animals (2 NMJs per animal) were examined per genotype for each criteria quantified. For analysis of axonal transport, kymographs were generated and analyzed from a total of 60 axons from 10 individual animals. For electrophysiology, for each genotype we analyzed 35 individual EJP traces and a 45s-long mEJP trace per muscle for at least 11 muscles (derived from at least 6 independent animals – 2 muscles per animal).

Data was analyzed by either Student's t-test (two groups) or one-way ANOVA followed by Tukey test (multiple groups). Normality of datasets for parametric/ANOVA was confirmed using the D'Agostino-Pearson-omnibus K2 test. p values smaller than 0.05 were considered statistically significant. All p values are indicated as *p<0.05, **p<0.01, and ***p<0.001 and ****p<0.0001. Data are presented as mean ±SEM.

## Acknowledgements

We thank Thomas Schwarz, Noreen Reist and Aaron DiAntonio for sharing antibody and *Drosophila* reagents, Dave Featherstone, Crystal Miller and Heather Broihier for tips and guidance on embryo studies, Mostafa Ghannad-Rezaie for help with kymograph analysis, Joeph Knoedler, Samhitha Raj and Chen Zhang, for helpful technical discussions, Raphael Zinser, Jeannine Kern, Christian, and Dhwani Joshi for technical support, Dion Dickman, Pragya Goel, Aaron DiAntonio and Alexandra Russo for discussion of unpublished electrophysiology data, and Robert Kittel, Ryan Insolera, Laura Smithson, and Dhananjay Yellajoshyula for discussions of data and paper drafts. Richard Daniels generated and shared *vglut*-DsRed fly lines and gave helpful technical advice for electrophysiology. This work was supported by a grant from the National Institute of Health (NS069844) to CAC and the Deutsche Forschungsgemeinschaft (DFG, RA 1804/2-1) to TMR In addition the Chica and Heinz Schaller Foundation (CHS) to TRJ supported TMR.

## Additional information

### Funding

| Funder | Grant reference number | Author |
|---|---|---|
| National Institutes of Health | RO1 NS069844 | Catherine A Collins |
| Deutsche Forschungsgemeinschaft | RA 1804/2-1 | Tobias Rasse |
| Chica and Heinz Schaller Foundation | | Thomas Robert Jahn |

The funders had no role in study design, data collection and interpretation, or the decision to submit the work for publication.

### Author contributions

Jiaxing Li, Conceptualization, Data curation, Formal analysis, Validation, Investigation, Methodology, Writing—original draft, Writing—review and editing; Yao V Zhang, Conceptualization, Data curation, Formal analysis, Funding acquisition, Validation, Investigation, Methodology, Writing—review and editing; Elham Asghari Adib, Formal analysis, Investigation, Methodology; Doychin T Stanchev, Formal analysis, Validation, Investigation, Methodology; Xin Xiong, Conceptualization, Data curation, Investigation, Methodology, Writing—review and editing; Susan Klinedinst, Conceptualization, Investigation, Writing—review and editing; Pushpanjali Soppina, Investigation, Methodology; Thomas Robert Jahn, Discussion of data, Resources and Funding acquisition for Tobias Rasse; Richard I Hume, Resources, Formal analysis, Supervision, Methodology, Writing—review and editing; Tobias M Rasse, Conceptualization, Resources, Data curation, Supervision, Funding acquisition, Investigation, Methodology, Writing—original draft, Project administration, Writing—review and editing, writing - conceptualization and final paper; Catherine A Collins, Conceptualization, Resources, Data curation, Supervision, Funding acquisition, Investigation, Methodology, Writing—original draft, Project administration, Writing—review and editing

### Author ORCIDs

Thomas Robert Jahn [ID] https://orcid.org/0000-0002-9266-6736
Catherine A Collins [ID] http://orcid.org/0000-0002-1608-6692

### Decision letter and Author response

Decision letter https://doi.org/10.7554/eLife.24271.031
Author response https://doi.org/10.7554/eLife.24271.032

## Additional files

### Supplementary files

• Transparent reporting form
DOI: https://doi.org/10.7554/eLife.24271.030

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
