## [Decision Letter]

Thank you for submitting your work entitled "The Wallenda/DLK kinase restrains the expression of pre-synaptic proteins according to their transport" for consideration by *eLife*. Your article has been reviewed by three peer reviewers, and the evaluation has been overseen by a Reviewing Editor and a Senior Editor. The reviewers have opted to remain anonymous.

Li et al. present work investigating Wnd/DLK MAP kinase pathway regulation of expression levels of synaptic genes following disrupted axonal trafficking upon knockdown of Unc-104/Kif1a, a Kinesin 3 motor. Based on the exciting finding that loss of wnd suppresses the reduced presynaptic active zone (AZ) protein accumulation and associated functional deficits of unc-104 hypomorphic mutants, the authors propose a model in which the Wnd pathway 'senses' levels of presynaptic proteins in the cell body and acts as a brake on expression when levels exceed the ability of the cell to transport these proteins to synapses. The experiments are generally carefully done and individual experiments serve the purpose of supporting the genetic interaction. However, as a whole, the paper worryingly feels like a catalog of correlative connections, missing conceptual depth. A consistent term used throughout the paper is "modest effect" or "modest role", suggesting ambiguity of the observations and the conclusions. Some of the issues could be addressed by re-writing, others need additional data. Thus, while this is an interesting model, it is not fully supported by the data presented as detailed below.

While some of our concerns can be addressed by better analysis and writing, others require significant experiments. We had prolonged discussions around whether these concerns can be addressed in two months and if done would we be persuaded. We feel that some suggested experiments which are critical to the thrust of the paper can be done in two months. Yet, when we asked if these concerns were addressed whether the model proposed and data presented would be acceptable despite 'mechanistic' gaps, we were not persuaded. In this situation, another round of review would not be correct.

*Reviewer #1:*

Li et al. present work investigating Wnd/DLK MAP kinase pathway regulation of expression levels of synaptic genes following disrupted axonal trafficking upon knockdown of Unc-104/Kif1a, a Kinesin 3 motor, and during development. Based on the exciting finding that loss of wnd suppresses the reduced presynaptic active zone (AZ) protein accumulation and associated functional deficits of unc-104 hypomorphic mutants, the authors propose a model in which the Wnd pathway 'senses' levels of presynaptic proteins in the cell body and acts as a brake on expression when levels exceed the ability of the cell to transport these proteins to synapses. While this is an interesting model, it is not fully supported by the data presented as detailed below.

1) Support for their model:

A) The authors propose that unc-104 function can be divided into a transport role for SVs that is independent of Wnd and a Wnd/DLK pathway-dependent role in regulating AZ proteins. The differentiation between Brp and SV proteins is based on the finding that reduced synaptic levels of Brp in unc-104 nulls are suppressed by wnd but SV protein levels are not, in contrast to their findings in unc-104 hypomorphs. Simpler models should be considered. For example, the observation that wnd suppresses the loss of SV proteins at synapses in hypomorphs with residual Unc-104 activity but not nulls is more easily explained by the model that SV transport, but not Brp transport, is fully dependent on Unc-104, consistent with prior observations in *Drosophila* (Pack-Chung et al., 2007)..

B) The strongest evidence for activation of Wnd in Unc-104 mutants comes from the puc-LacZ experiments in unc-104 RNAi lines (Figure 3). Given the importance of this finding, it this experiment should be conducted in unc-104 null and hypomorphic mutants. It's not clear how the proposed model leads to the specificity for unc-104 reported here. Have the authors investigated if cell body levels of presynaptic proteins are increased in these other mutants?

In Figure 8, the authors show that GFP-tagged Wnd MiMIC lines show increased Wnd in unc-104 mutants when well-characterized antibodies to Wnd (Collins et al., 2006) fail to show an increase. If they are going to include this data, the authors should provide an explanation for why the MiMIC data might provide a better indication of the underlying biology.

C) The evidence that Wnd regulates the levels of presynaptic protein levels and in particular that it acts at the level of transcription is somewhat inconsistent. Upon Wnd overexpression, the authors observed decreased protein levels at active zones as predicted. However, in wnd null embryos they see increased levels in some proteins at some stages, but decreases in other proteins (e.g., Brp). This and the absence of a consistent effect on the timing or expression, weakens their conclusion that wnd controls the levels and timing of expression of AZ and SV components.

In Figure 6—figure supplement 2, the authors present RT-PCR data that shows no decrease in transcription of synaptic proteins in unc-104 mutants as predicted by their model.

Other major concerns:

1) Much of the wnd, unc-104 interaction data was obtained using a previously published hypomorphic alleles of unc-104. Given their central role in this work, the authors should provide some information on the nature of these alleles. In the analysis of interactions between null mutants presented in Figure 5 and Figure 6, it is important to include an analysis of wnd mutants alone (as in 6C-G) to rule out additive effects. Have the authors analyzed dominant suppression of unc-104 phenotypes by wnd, which would strongly support the conclusion that the two genes function in a linear pathway?

2) An electrophysiological analysis was conducted in hiw mutants (where Wnd levels are increased), but should be completed in wnd overexpression lines to more directly test the hypothesis that increased Wnd explains functional defects in unc-104 mutants. Here, the authors present the surprising result that wnd suppresses the physiological defect in hiw mutants. This is in contrast to results published in an earlier paper with overlapping authors (Collins et al., 2006). They suggest the difference may be due to the use of different wnd alleles, but this needs to be addressed experimentally given the importance of wnd suppression phenotypes to this work and the high impact of the previously published result that is called into question.

3) Given that synaptic function is fully suppressed in hypomorphs, it is surprising that locomotion and viability show little improvement. This disconnect should be further addressed by evaluating the relevant allelic combinations, not just RNAi. Relatedly, how is there so much movement in double null mutants (Figure 5—figure supplement 1) when there is essentially no SV transport to AZs?

4) It was cumbersome to parse through the data as presented in 21 figures (9 main plus 12 supplemental) including 2 separate model figures. Figure 6 was presented in a way that made it especially difficult to compare genotypes. It also looks like the effect of wnd loss of function on Brp levels is very different between Figure 1 (no effect) and 4G (nearly doubled).

*Reviewer #2:*

In "The Wallenda/DLK kinase restrains the expression of pre-synaptic proteins according to their Transport" Drs. Collins, Rasse and colleagues reveal an unexpected novel role for Unc-104 in regulating synaptic development and bouton composition in a manner not directly related to its long-range transport functions.

Synaptic vesicle precursor proteins and active zone proteins are synthesized in neuronal cell bodies and transported to synapses. Kinesin motors (Unc-104) transport SV precursors. AZ components are reduced at the synaptic terminals of the unc-104 mutants, but their transport has never been shown to be driven directly by unc-104. Through an exhaustive set of experiments (using diverse mutant and RNAi combinations), the authors show that defects in unc-104 lead to the activation of the Wnd signaling pathway (via Bsk and Fos), which in a cell-autonomous manner regulates AZ protein levels in neuronal cell bodies and synaptic terminals. Suppressing Wnd signaling in unc-104 strong hypomorphs or in RNAi knockdowns significantly reduces the defects in AZ at the synapse, defects in synaptic transmission, larval mobility and viability, but only partly rescues SV protein levels. Furthermore, puckered expression and axonal regeneration, both of which require enhanced Wnd signaling, are enhanced in unc-104 mutants.

Crucially, through genetic interaction studies with null mutants of unc-104 (as opposed to hypomorphs) the authors revealed two separable functions for unc-104: one for trafficking SV associated proteins to the synapse (VGlut1, Syt1, Csp) and one for activating the Wnd pathway, which in unc-104 mutants lead to the decrease of multiple synaptic proteins.

While these represent important, transformative, and convincing advances in our understanding of unc-104 and Wnd/DLK signaling in synapse formation, the one weakness of the paper is that it does not uncover HOW Unc-104 regulates Wnd signaling, or how Wnd signaling constrains synaptic protein levels. Filling this gap is likely beyond the scope of the current work and not necessary to merit publication in *eLife*, but there are several points that the authors could address based on their current approaches:

1) It is important for the authors to quantify Wnd-MiMIC levels in the neuronal cell bodies (akin to the way they quantified UAS-WnkKD in supplemental Figure 1 to Figure 8). They should also explain why they used the kinase dead version and whether they believe it behaves differently from wild type Wnd.

2) The authors present hiw data but do not address hiw function in their model or discussion. The authors should measure hiw RNA and/or protein levels in Unc-104 mutants to ask if hiw is contributing to the phenotype. If hiw protein (but not RNA) is downregulated in the unc-104 mutant, it might suggest that autophagy-mediated degradation of Hiw (as per Shen et al., 2009), and perhaps other synaptic proteins, underlies the regulatory mechanism by which unc-104 constrains synaptic protein levels in response to cell body accumulation of cargo. This could be simply tested by evaluating the amount and distribution of an autophagy flux marker such as ATG8-GFP-mCherry.

3) The observation that ectopic overexpression of Brp, SytI and VGlut leads to increased puc-lacZ reporter signal begs the question whether this also triggers changes in Wnd (or hiw) levels or localization, as these results could provide more insight into the mechanism via which Unc-104 deficits lead to Wnd activation.

*Reviewer #3:*

This paper describes a genetic interaction between the Unc-104 motor protein and the Wnd DLK kinase. The authors presented impressive amount of data from synapse cell biology to physiology with an attempt to conclude that Wnd activation is sensitive to disruption of unc-104 mediated transport of some axonal components. The measurement for Wnd activity mostly relied on induction of a puc-LacZ reporter in the soma, and morphological changes of NMJ terminals. While experiments are generally carefully done, individual experiment serves the purpose of supporting the genetic interaction. However, as a whole, the paper feels like a catalog of correlative connections, missing conceptual depth. A consistent term used throughout the paper is "modest effect" or "modest role", suggesting ambiguity of the observations and the conclusions. Some of the issues could be addressed by re-writing, others need additional data.

Their observation that Wnd suppresses the lethality of unc-104 mutants is the key observation led to this study. But it is not clear whether this suppression is directly caused at the level of presynaptic transport. Does inhibition of Wnd downstream factors, *JNK* and FOS, also suppress unc-104 lethality? In subsection “Wnd inhibits synapse assembly independently of cargo transport downstream of Unc-104” the final sentence of the second paragraph seems to state that "most or all" unc-104; wnd mutants actually die. Is the suppression of lethality rather minor?

In axon regeneration analysis in unc-104 mutants, the observed effects should be rescued using wild type unc-104.

Subsection “Activation of the Wnd signaling pathway in neurons is sufficient to impair presynaptic structure and synaptic transmission”: first paragraph, the authors state "ectopic activation of Wnd should mimic the unc-104 mutant phenotypes". But where is Wnd normally activated?

In the same section, second paragraph: The authors made a comment about the result in Figure 4 being differed from their previous study that used a hypomorphic allele of Wnd". They should perform the same analysis in parallel on both null and that hypomorphic allele.

They talked about similarities of how loss of Hiw or of Unc-104 affect Wnd function. An important experiment should include a double mutant study of Hiw and unc-104.

[Editors’ note: what now follows is the decision letter after the authors submitted for further consideration.]

Thank you for choosing to send your work entitled "The Wallenda/DLK kinase restrains the expression of pre-synaptic proteins according to their transport" for consideration at *eLife*. Your letter of appeal has been considered by a Senior Editor in discussion with the reviewers, and we are prepared to consider a revised submission with no guarantees of acceptance.

While we are happy to consider a revised manuscript, you will see that we have some serious concerns that need careful attention. We see merit in the author's rebuttal letter that the most concerns can be addressed with rewriting and some additional experiments and key issues addressed. The overall impact of the work, if these points are well-addressed, can be strong, since it provides a fundamentally new model for how unc-104 contributes to neuronal function, by activating an axonal injury pathway. However, the data and model also need to be explained more clearly to be digestible by a broad readership such as at *eLife*.

In response to the concerns raised about inconsistent results in their wnd loss-of-function analysis, the authors are dismissive, imploring us to "Note that our model is NOT that Wnd functions as a major restraint on these proteins during normal conditions or during development," yet their subheading describing these results is "Wnd restrains the expression of presynaptic proteins at early stages of synaptogenesis." In a second example, the authors dismiss concerns about their analysis of MiMIC-GFP instead of endogenous Wnd levels by simply stating that the Wnd antibody does not work well for immunohistochemistry without addressing the fact that they have previously published analogous experiments using their Wnd antibody for immunohistochemistry (Collins et al., 2006). The authors may want to pay particular attention to these points.

The authors should measure puc-LacZ expression in additional unc-104 mutants. Also, rescue of additional phenotypes aside from NMJ morphology and active zone apposition to PSD (e.g. lethality or synaptic function) by BSK and FOS would provide strong support for the model. However, I was not concerned that the Mimic tool showed an increase in Wnd while the western blot did not, since the Mimic tool accounts for changes in subcellular distribution – this point can be addressed better by the authors in the manuscript. However, it is absolutely essential that the MIMIC phenotype be quantified and use of the Kinase Dead explained in the text.

The effect of wnd on the timing and expression of presynaptic proteins needs to be much more clearly explained, as discussed in the rebuttal. As it stands this section is very distracting and confusing. The authors should also clarify their model for how synaptic protein levels are regulated in the unc-104 mutants (though the authors do discuss the RT-PCR data and I agree with them that RT-PCR detects transcripts from all the cell types in all the tissues used for the mRNA prep, and does not account for specific defects in protein levels and localization at the neuronal cell bodies and synaptic terminals). The main point of the paper does not hinge on a particular model of synaptic protein level control, but the authors do need to clarify their explanations, either in the text or with more experiments.

Wnd null mutants alone should be included in Figure 5 and Figure 6 for comparison in this specific assay. However, though phenotypes in transheterozygotes can be informative, lack of a dominant transheterozygote phenotype might just reflect gene dose requirements, and so we did not put as much weight on that particular experiment.

Electrophysiological analysis (and synaptic vesicle protein levels) in wnd overexpression lines would provide strong corroboration that increased Wnd explains functional defects in unc-104 mutants. Addressing the discrepancy with previous results (Collins 2006) on Wnd suppression of the hiw phenotype will be also be important.

On the other hand, experiments to address the cellular function of hiw would be nice but are not essential to the main interesting points of the paper. If the authors have the data or a more clear model of the role of hiw, it would be helpful to include, but we don't think it's necessary.

The rescue of the axon regrowth phenotype are not critical for the main conclusions of the paper and could take a lot of time.

[Editors' note: further revisions were requested prior to acceptance, as described below.]

Thank you for submitting your article "Restraint of presynaptic protein levels by Wnd/DLK signaling mediates synaptic defects associated with the kinesin-3 motor Unc-104" for consideration by *eLife*. Your article has been reviewed by two peer reviewers, and the evaluation has been overseen by K VijayRaghavan as the Senior Editor and the Reviewing Editor. The reviewers have opted to remain anonymous.

The reviewers have discussed the reviews with one another and the Reviewing Editor has drafted this decision to help you prepare a revised submission.

As you can see your manuscript needs only a few minor points that need to be addressed. Please do that speedily. We can then accept the revised manuscript after an editorial examination.

*Reviewer #1:*

In their revised manuscript, the authors have been very responsive to reviewer suggestions. The logical flow of the manuscript is significantly improved and experimental concerns directly addressed.

I have one question regarding the addition of wnd null data to Figure 1 and Figure 7—figure supplement 1. Is it still accurate to state that "Control and Experimental animals were always dissected, fixed and stained in the same dish/slide and imaged in parallel using identical confocal settings" (Experimental Procedures,), or were the new data obtained in separate experiments and normalized to wild type for incorporation with previously obtained data?

*Reviewer #2:*

The manuscript is much improved with the suggested experiments, reorganization, and rewriting. My concerns have been adequately addressed and I think that this is a very interesting, well-executed, and thought-provoking study.

---

## [Author Response]

[Editors’ note: the author responses to the first round of peer review follow.]

Reviewer #1:

Li et al. present work investigating Wnd/DLK MAP kinase pathway regulation of expression levels of synaptic genes following disrupted axonal trafficking upon knockdown of Unc-104/Kif1a, a Kinesin 3 motor, and during development. Based on the exciting finding that loss of wnd suppresses the reduced presynaptic active zone (AZ) protein accumulation and associated functional deficits of unc-104 hypomorphic mutants, the authors propose a model in which the Wnd pathway 'senses' levels of presynaptic proteins in the cell body and acts as a brake on expression when levels exceed the ability of the cell to transport these proteins to synapses. While this is an interesting model, it is not fully supported by the data presented as detailed below.

We appreciate that the reviewer recognizes the novelty and strength of our findings and the model we propose. With a careful and much appreciated eye on the data, the reviewer summarizes below some concerns about individual observations. We are optimistic that our explanations and clarifications described here, along with some adjustments and clarifications in the writing, can resolve this reviewer’s concerns.

1) Support for their model:A) The authors propose that unc-104 function can be divided into a transport role for SVs that is independent of Wnd and a Wnd/DLK pathway-dependent role in regulating AZ proteins.

This statement is not precise for what we propose. Please see the clarifications below.

The differentiation between Brp and SV proteins is based on the finding that reduced synaptic levels of Brp in unc-104 nulls are suppressed by wnd but SV protein levels are not, in contrast to their findings in unc-104 hypomorphs.

To clarify, in *unc-104; wnd* double mutants compared to *unc-104* mutants alone, we see a major increase in total levels (across the sum of the cell body, axon and synaptic compartments) for *both* AZ and SV proteins. This is evident for both *unc-104-*hypomorph (Figure 6, Figure 6—figure supplement 1 C) and null mutations (Figure 5, Figure 6, & Figure 6—figure supplement 1 A-B). These observations lead to our major interpretation that the Wnd pathway, which becomes activated in *all of the unc-104 mutants*, down-regulates the levels of *both AZ and SV* proteins.

Simpler models should be considered. For example, the observation that wnd suppresses the loss of SV proteins at synapses in hypomorphs with residual Unc-104 activity but not nulls is more easily explained by the model that SV transport, but not Brp transport, is fully dependent on Unc-104, consistent with prior observations in Drosophila (Pack-Chung et al., 2007).

This explanation is indeed incorporated within our model (Figure 9). This explanation accounts for the suppression of AZ but not SV localization in the *unc-104* mutants. However it is not the full picture, since this explanation does not account buildup of both AZ and SV components in the cell bodies of *unc-104-null; wnd* and *unc-104-hypomorph;wnd* double mutants (Figure 6).

The most important part of the model is that the Wnd pathway, which becomes activated when unc-104 function is lost, promotes a decrease the total cellular levels of many AZ and SV proteins. This effect is an important element of the synaptic phenotypes in both *unc-104-hypomorph* and *null* mutants. Our findings with *unc-104-null* mutants in Figure 5 and Figure 6 show that this effect for Wnd signaling can be genetically and mechanistically separated from a transport role of Unc-104.

We originally interpreted that the reviewer may be cautioning us to be careful about implying precise motor-cargo relationships. It is indeed possible that Unc-104 plays some partial role in AZ localization – we simply need to be precise in our language. However our primary point is not about which motor carries which cargo. Instead, our findings explicitly reveal the pitfalls to interpreting motor cargo relationships from standard phenotypic analysis. Our finding that an axonal injury pathway is activated in *unc-104* mutants provides a new view with alternative interpretations for other published studies in the field. Several relevant previous observations along with the exciting implications are discussed in Discussion section.

B) The strongest evidence for activation of Wnd in Unc-104 mutants comes from the puc-LacZ experiments in unc-104 RNAi lines (Figure 3). Given the importance of this finding, it this experiment should be conducted in unc-104 null and hypomorphic mutants.

We agree that evidence for Wnd’s activation in the *unc-104* mutants is an important part of the paper. We show this for two independent RNAi lines, which show similar phenotypes to the multiple hypomorphic alleles (*bris* and *O3.1*) used in this paper across other assays. The suggestion to show this for additional alleles is do-able. We already have *puc*-lacZ combined into *unc-104*-hypomorph lines so we think it would be straight-forward to include this additional data.

Importantly, we already present additional support that the Wnd pathway is activated. In *unc-104-hypomorph* mutant and RNAi lines we see synaptic over-branching phenotypes that are hallmark of Wnd activation, and these are dependent upon Wnd and downstream *JNK* and Fos signaling components (Figure 3—figure supplement 3, and also data not shown). We are thinking to move this data to the main part of Figure 3 instead of supplemental data for the revision.

We also show in hypmorph/null mutants that Wnd protein is physically altered in levels and localization by immunohistochemistry using two independent reagents: the GFP-tagged MiMIC-Wnd line (Figure 8) which tags endogenous Wnd with GFP, and also for transgenically expressed GFP-Wnd-KD (Figure 8—figure supplement 1), which has been used in previous studies of Wnd’s regulation (Collins et al., 2006; Xiong et al., 2010; Yan et al., 2016).

And finally, our epistasis experiments with *unc-104* mutants suggests strongly that the Wnd pathway is responsible for synaptic phenotypes in *unc-104* mutants, hence functions downstream. This analysis in late-staged embryos is technically challenging – only a few labs in the world can do it and it represents 9 months of effort on our part. The results were important for reaching an understanding of the relationship.

It's not clear how the proposed model leads to the specificity for unc-104 reported here. Have the authors investigated if cell body levels of presynaptic proteins are increased in these other mutants?

We see cell body accumulations of presynaptic proteins in multiple hypomorph and null *unc-104* mutant combinations (Figure 6: *bris/P350* and *P350/P350*; Unpublished: *bris/d11204; o3.1/P350; 170/170; 52/52*), and also for *unc-104* RNAi knockdown in neurons (pan-motoneurons, using OK6-Gal4 or single motoneuron, using m12-Gal4). More importantly, under all these different conditions, when activation of the Wnd signaling is reversed/suppressed via wnd RNAi-knockdown or in *wnd, unc-104* double mutants, the cell body accumulation is dramatically enhanced.

Upon reviewing the manuscript after reading the reviewer’s queries, we realize it would be helpful to provide more background information about the different ‘hypomorph’ alleles, and clarify/explain in more detail how all of the alleles show similar synaptic phenotypes and relationships with Wnd to that of simple knock-down of Unc-104 with RNAi.

In contrast to the multiple mutations that reduce Unc-104 function, mutations that impair kinesin-1 function show accumulations of presynaptic proteins within segmental nerves rather than in the cell bodies, and also do not significantly induce the puc-lacZ reporter (Figure 3).

In Figure 8, the authors show that GFP-tagged Wnd MiMIC lines show increased Wnd in unc-104 mutants when well-characterized antibodies to Wnd (Collins et al., 2006) fail to show an increase. If they are going to include this data, the authors should provide an explanation for why the MiMIC data might provide a better indication of the underlying biology.

To examine Wnd protein levels, we employed multiple approaches (Figure 8). The Wnd antibody works great for Western Blots but not well for immunohistochemistry. The GFP epitope within the MiMIC-GFP-Wnd line allows for some detection by immunohistochemistry (which we show controls for here), hence an examination of protein localization. We were not able to detect a global increase in the levels of Wnd protein within brain extracts by Western Blot, but we were able to see changes in the enrichment of MiMIC-GFP-Wnd in cell bodies (but not in axons and synapses). In addition, we observe changes in the levels and localization of transgenically expressed GFP-Wnd-KD (Figure 8—figure supplement 1), which has been used in previous studies of Wnd’s regulation (Collins et al., 2006; Xiong et al., 2010; Yan et al., 2016).

The observation that a global increase in endogenous Wnd levels was not detected by Western Blot may benefit from some further clarification and discussion. The Western Blots are necessarily carried out with extracts made from dissected larval CNS/brains, pooled in entirety. We cannot isolate individual neurons for this analysis. We’ve observed that the activation of Wnd and the regulation of presynaptic proteins by Wnd appears most striking in motoneurons. The potential specificity to motoneurons makes it harder to detect simple biochemical evidence for the relationship using standard biochemical methods from whole larval brains.

C) The evidence that Wnd regulates the levels of presynaptic protein levels and in particular that it acts at the level of transcription is somewhat inconsistent. Upon Wnd overexpression, the authors observed decreased protein levels at active zones as predicted. However, in wnd null embryos they see increased levels in some proteins at some stages, but decreases in other proteins (e.g., Brp). This and the absence of a consistent effect on the timing or expression, weakens their conclusion that wnd controls the levels and timing of expression of AZ and SV components.

An important but overlooked point here is that Wnd signaling is normally highly restrained in wild type, uninjured animals. If its role is primarily to respond to injury or stress, then it is expected that *loss-of-function* phenotypes in an uninjured background should be mild. With an original motivation to understand whether there is also a role for Wnd during normal synaptic development, we examined these phenotypes carefully during development. We found that the *wnd loss-of-function* phenotypes are indeed mild. However, strikingly, we did notice some elevation in VGlut and Syt protein co-incident with the onset of expression for these proteins. Note that our model is NOT that Wnd functions as a major restraint on these proteins during normal conditions or during development. Rather, Wnd restrains these proteins when it is substantially activated. In normal conditions this may occur only at extremely low levels to keep total synaptic protein levels in balance.

The fact that we did not observe an increase in Brp does not negate the model. Very little is currently known about how the expression of AZ and SV proteins are regulated developmentally –this is an interesting topic for future studies. It is likely that multiple mechanisms exist for different proteins, and that Brp levels are not sufficiently high during these stages of normal development to detect a role for Wnd in uninjured animals.

In Figure 6—figure supplement 2, the authors present RT-PCR data that shows no decrease in transcription of synaptic proteins in unc-104 mutants as predicted by their model.

As discussed above for point B, studies with whole brain extracts (for Western Blot or RT-PCR analysis, referenced here) are likely to miss the cell specific regulation which is occurring in motoneurons. It is a future direction to profile the responses in motoneurons specificially. This requires some additional technical innovations since cell sorting causes axotomy, which activates the Wnd pathway on its own.

Other major concerns:1) Much of the wnd, unc-104 interaction data was obtained using a previously published hypomorphic alleles of unc-104. Given their central role in this work, the authors should provide some information on the nature of these alleles.

We agree and can provide further clarification in the text. An important clarification point is that we see similar suppression by *wnd* mutants for multiple hypomorphic allele combinations of *unc-104* mutants (See point B above and subsection “The Wnd signaling pathway mediates presynaptic defects in unc-104 mutants”), and also when Unc-104 protein levels are simply reduced by RNAi-knockdown. So we think that any manipulation that impairs Unc-104’s basic function leads to activation of Wnd signaling.

In the analysis of interactions between null mutants presented in Figure 5 and Figure 6, it is important to include an analysis of wnd mutants alone (as in 6C-G) to rule out additive effects.

We have indeed included detailed analysis of *wnd* mutants alone at the embryonic stages in Figure 7. We should be able to include for reference some of the relevant data Figure 6 for comparison.

Have the authors analyzed dominant suppression of unc-104 phenotypes by wnd, which would strongly support the conclusion that the two genes function in a linear pathway?

We’ve observed that partial knockdown of Wnd signaling by RNAi can suppress *unc-104* phenotypes (Figure 1—figure supplement 1B). We feel that experiments with *wnd* heterozyogotes would not really add much more to this paper, especially since it is not clear in our data that removal of one copy of *wnd* leads to a 50% reduction of Wnd levels in all cell types.

The reviewer may instead be referring to double-heterozygote experiments, which can be helpful for revealing relationships in a linear pathway via their sensitivity to gene dosage of both factors. This kind of analysis is more do-able when both genes have similar loss-of-function phenotypes. However *unc-104* and *wnd* mutants have opposite *loss-of-function* phenotypes, so we’re not sure what we would learn from a double heterozygote.

2) An electrophysiological analysis was conducted in hiw mutants (where Wnd levels are increased), but should be completed in wnd overexpression lines to more directly test the hypothesis that increased Wnd explains functional defects in unc-104 mutants. Here, the authors present the surprising result that wnd suppresses the physiological defect in hiw mutants. This is in contrast to results published in an earlier paper with overlapping authors (Collins et al., 2006). They suggest the difference may be due to the use of different wnd alleles, but this needs to be addressed experimentally given the importance of wnd suppression phenotypes to this work and the high impact of the previously published result that is called into question.

The main point to communicate in Figure 4 is that ectopic activation of Wnd alone is sufficient to induce defects in synapse structure and function. We think that the suggested physiology characterization for *wnd* over-expression could further and more simply support this point. These experiments should be complete-able in a relatively short time frame, so we will aim to add this data to the revised manuscript. If we can indeed include this data, then we may then want to move the *hiw* data to supplemental data, where the partial differences with the 2006 study are less distracting.

While *hiw* mutants (and *unc-104* mutants) show mild reductions in quantal content, they are significantly reduced for mini frequency. The observation that *wnd* mutations suppress *hiw*’s defect in mini frequency are actually conserved between both our observations and the previous 2006 study. The single difference is that the 2006 study did not reveal a suppression of the defect in quantal content, while ours does. We would indeed like to understand this better – resolving the differences will require repeating exactly the same allele combinations in our own conditions for comparison. This is do-able and we are willing to do it. The main point of this would be to provide further context for previous observations in the field rather than to further validate the observations we have made here in this paper (which we see for two allelic combinations and are consistent with the morphological data).

3) Given that synaptic function is fully suppressed in hypomorphs, it is surprising that locomotion and viability show little improvement. This disconnect should be further addressed by evaluating the relevant allelic combinations, not just RNAi. Relatedly, how is there so much movement in double null mutants (Figure 5—figure supplement 1) when there is essentially no SV transport to AZs?

Actually, neither of these results are surprising.

Our study indicates that the effects of Wnd signaling (which is activated in *unc-104* mutants and is responsible for many interesting synaptic phenotypes) are separable from a primary function for Unc-104 in transporting SVs (Figure 5). In addition, Unc-104/KIF1A has many other proposed roles, including regulation of autophagy (Stavoe et al., 2016), lysosome transport (Guardia et al., 2016), insulin secretion (Cao et al., 2014), and cell migration (Carabalona et al., 2016). It would be presumptuous to assume or expect that we could completely rescue lethality in these animals, which are quite defective in many aspects. Likewise for locomotion: while we see strong phenotypes at the NMJ synapse, this is not the only synapse that animals need for locomotion. It is remarkable that we see ANY improvement in locomotion and viability.

For the *unc-104-null* mutants, the rescue is also extremely mild and we state this in the text (Subsection “Wnd inhibits synapse assembly independently of cargo transport downstream of Unc-104”). Perhaps the strong difference in bar graphs in Figure 5—figure supplement 1 give an impression there is a stronger suppression. However note that this is simply the “percentage of animals showing muscle movement” not the actual ‘degree’ of movement, and the movement is observed only immediately after the animals hatch. They die soon thereafter.

We worry based on these points and also the summary statement that these ‘modest’ or ‘inconsistent’ effects upon lethality and motility were a primary criticism for the paper. Therefore it is important to clarify that these effects are consistent with the model we present in the paper (if they were stronger they would actually argue against it). The ‘modest’ rescue of lethality should not diminish the importance of our findings, which identify specific cellular roles for Wnd in *unc-104* mutants.

4) It was cumbersome to parse through the data as presented in 21 figures (9 main plus 12 supplemental) including 2 separate model figures.

We appreciate the efforts from the reviewer to evaluate all of the data, and their feedback has helped us see how we might be able to improve the writing to make the paper less cumbersome to read. It has been over 7 years of hard efforts between our two labs to reach our current understanding of the surprising relationship between Unc-104 and Wnd. There are many more data that are not included in this paper. We will work further on the writing to better communicate how the individual experiments build upon one another to fit into the big picture story.

Figure 6 was presented in a way that made it especially difficult to compare genotypes.

We appreciate this point. It is important to consider the levels in the different cellular compartments and to assess differences in total levels accounting for all of the compartments. This comparison is our goal for Figure 6. If one examines synaptic levels alone (in Figure 1), that is only part of the story.

It also looks like the effect of wnd loss of function on Brp levels is very different between Figure 1 (no effect) and 4G (nearly doubled).

The *wnd* mutants in Figure 1 (1.4 fold) and 4G (2 fold) are slightly different in genotype and strain background (*wnd^3/3^* and *wnd^3/df^,* respectively). Since the *wnd^3/3^* animals are maintained in the same strain background there is a chance to accumulate genetic modifiers of the phenotype in the background. We prefer to carry out all of our studies with two different alleles crossed to one another to be certain the phenotype is indeed due to the mutation we are manipulating.

Reviewer #2:

[…] While these represent important, transformative, and convincing advances in our understanding of unc-104 and Wnd/DLK signaling in synapse formation, the one weakness of the paper is that it does not uncover HOW Unc-104 regulates Wnd signaling, or how Wnd signaling constrains synaptic protein levels. Filling this gap is likely beyond the scope of the current work and not necessary to merit publication in eLife, but there are several points that the authors could address based on their current approaches:

We are glad that reviewer 2 shares the same point of view with us in both strengths and one weakness. We also agree that this weakness is beyond the scope and not necessary for our current work to be published. The reviewer’s suggestions listed below are fair and we are ready to address all of these points.

1) It is important for the authors to quantify Wnd-MiMIC levels in the neuronal cell bodies (akin to the way they quantified UAS-WnkKD in supplemental Figure 1 to Figure 8). They should also explain why they used the kinase dead version and whether they believe it behaves differently from wild type Wnd.

The quantification of Wnd-MiMIC levels can certainly be included in the revised manuscript. Ectopic expression of Wnd (lacking the kinase mutation) leads to lethality and technical complications. We can also add additional information about the Wnd-kinase-dead (KD) transgene, which has been used in previous studies to uncover biologically important mechanisms of regulation (Collins et al., 2006; Xiong et al., 2010; Yan et al., 2016). See also our discussion to reviewer 1’s point B, above.

2) The authors present hiw data but do not address hiw function in their model or discussion. The authors should measure hiw RNA and/or protein levels in Unc-104 mutants to ask if hiw is contributing to the phenotype. If hiw protein (but not RNA) is downregulated in the unc-104 mutant, it might suggest that autophagy-mediated degradation of Hiw (as per Shen et al., 2009), and perhaps other synaptic proteins, underlies the regulatory mechanism by which unc-104 constrains synaptic protein levels in response to cell body accumulation of cargo. This could be simply tested by evaluating the amount and distribution of an autophagy flux marker such as ATG8-GFP-mCherry.

These interesting suggestions center around the goal of understanding whether Hiw, a major known regulator of Wnd, plays a role in Wnd’s activation in *unc-104* mutants. Based on several observations thus far (some in the manuscript and some yet unpublished), we think that loss of Hiw function is unlikely to explain or mediate the activation of Wnd in *unc-104* mutants.

1) The effects of *hiw* and *unc-104* mutants upon the levels and localization of Wnd are strikingly different: in *hiw* mutants, the levels of Wnd are strongly elevated throughout the animal and this elevated Wnd protein is observed to localizes to synapses and neuropil (Collins et al., 2006, and also Figure 8). In contrast, *unc-104* mutants cause more modest changes to Wnd protein and levels, and these changes are only observed in cell bodies, not synapses (Figure 8 and supplements).

*2) hiw* mutants show a strong delay in axonal degeneration (Xiong et al., 2012; Babetto et al., 2013). However we found no delay to degeneration in *unc-104* mutants (unpublished).

3) Western blot results did not show an obvious reduction in Hiw protein level when *unc-104* is knocked down (unpublished).

The potential role of autophagy is an interesting topic to further understand. Thus far we have found that null mutations in *atg7* and knock-down of atg13 or Fip200 in neurons, which are critical for different stages of autophagy (Fougeray and Pallet, 2015), did not alter Wnd signaling using *puc*-lacZ as a reporter (unpublished). This runs a bit counter to predications suggested by the Shen et al., 2009 study. So more needs to be done to sort this out and this is a topic for future work.

3) The observation that ectopic overexpression of Brp, SytI and VGlut leads to increased puc-lacZ reporter signal begs the question whether this also triggers changes in Wnd (or hiw) levels or localization, as these results could provide more insight into the mechanism via which Unc-104 deficits lead to Wnd activation.

This is a good suggestion but a negative or mild result could occur simply because the effect of overexpressing a single pre-synaptic protein does not activate the pathway to the same extent, so would not be useful in ruling out a model. We appreciate that the reviewer acknowledges that filling the gap of HOW Unc-104 regulates Wnd signaling ‘is likely beyond the scope of the current work and not necessary to merit publication in *eLife*’. We have some additional ideas on this question that we want to carry out in future studies.

Reviewer #3:

This paper describes a genetic interaction between the Unc-104 motor protein and the Wnd DLK kinase. The authors presented impressive amount of data from synapse cell biology to physiology with an attempt to conclude that Wnd activation is sensitive to disruption of unc-104 mediated transport of some axonal components.

This statement reflects some confusion between our major conclusions in the paper with models and ideas that we raise. We propose an idea that Wnd pathway activation is sensitive to build-up of Unc-104’s cargos. However this is the speculative idea in the paper, which is framed as a model for thinking about the data as a whole, combined with other observations in the field. It is not the primary goal of the paper to reach this idea as a ‘conclusion’.

The measurement for Wnd activity mostly relied on induction of a puc-LacZ reporter in the soma, and morphological changes of NMJ terminals.

In addition we showed an induction of Wnd protein levels and localization in motoneurons using two independent reagents/methods (endogenous MiMIC-tagged Wnd (Figure 8) and ectopically expressed GFP-Wnd (Figure 8—figure supplement 1), and extensive genetic interactions (Figure 1–Figure 3 and Figure 5 with accompanying supplements). Importantly, the genetic interactions include suppression of NMJ phenotypes in *unc-104* null mutants (Figure 5 and accompanying supplement), which suggests that Wnd functions downstream of Unc-104 to mediate major aspects of the synaptic phenotype. Please also see our discussion for reviewer 1 point B.

While experiments are generally carefully done, individual experiment serves the purpose of supporting the genetic interaction. However, as a whole, the paper feels like a catalog of correlative connections, missing conceptual depth. A consistent term used throughout the paper is "modest effect" or "modest role", suggesting ambiguity of the observations and the conclusions. Some of the issues could be addressed by re-writing, others need additional data.

This paragraph is also used in the summary statement. However the point that “the paper feels like a catalog of correlative connections, missing conceptual depth” is not well explained with sufficient points for us to understand how this conclusion is reached. The actual itemized points for this reviewer (as we discuss below) and the other reviews (as discussed above) are all readily addressable, and do not detract from the major conclusions and impact of this paper.

While the genetic interaction between Unc-104 and Wnd is very strong, there are indeed several results in the paper that show “modest” effects, and we also presented several negative data. However all of these data are helpful for reaching our understanding of the relationship between Unc-104 and Wnd. Importantly, all of these “modest” effects are consistent with the model we propose, and none are essential for the primary conclusions of the paper. Reasons for “modest effects are discussed in points C and 3 from reviewer 1. Due to the misunderstandings reflected in the comments, we plan in revision to do a better job with the writing to make these reasons clear up front, so that is easier for readers to follow the expectations for individual experiments and how they build upon one another.

Their observation that Wnd suppresses the lethality of unc-104 mutants is the key observation led to this study. But it is not clear whether this suppression is directly caused at the level of presynaptic transport. Does inhibition of Wnd downstream factors, JNK and FOS, also suppress unc-104 lethality? In subsection “Wnd inhibits synapse assembly independently of cargo transport downstream of Unc-104” the final sentence of the second paragraph seems to state that "most or all" unc-104; wnd mutants actually die. Is the suppression of lethality rather minor?

There is a significant misunderstanding here: the suppression of lethality data is NOT a ‘key observation’ in this study! Instead, the key observations that led to and are fundamental to our study are the striking suppression of synaptic phenotypes (for which we have extensive data in Figure 1, Figure 2, supplemental data and additional data not shown). We show in this paper specific cellular phenotypes that are mediated by Wnd, which provide a richer depth of information than is possible with simple lethality data.

The suppression of lethality is indeed only partial and this makes sense with the model we have reached. As we have detailed above in the discussion for reviewer 1’s point 3, the genetic data with *unc-104* null mutants tells us that while activated Wnd signaling is responsible for many of the synaptic phenotypes of *unc-104* mutants, this is separate from other important transport functions of Unc-104,

We suspect that some confusion about the expectations for the lethality data along with some confusion about its extent expressed by reviewer 1 may have distracted the reviewers’ discussion to result in the ultimately negative conclusion for this paper. With these points clarified we think the reviewers may reach a different conclusion.

In axon regeneration analysis in unc-104 mutants, the observed effects should be rescued using wild type unc-104.

While this experiment is do-able, it would not add much to the paper. We included the enhanced regeneration as further evidence that the Wnd pathway is activated in *unc-104* mutants. However we already have other data that more directly shows this point, so we could simply remove this data without impacting our conclusions for this paper. It may still be useful for the field have the regeneration data included for information that it may add to other published studies. For instance, a recent study from the Colon-Ramos lab (Stavoe et al., 2016) suggested that *unc-104* mutants show enhanced axonal outgrowth phenotypes in *C. elegans* PVD neurons. This may be consistent with the enhanced axonal regeneration phenotype we have observed here. However, because this is not essential data for the conclusions of our study, we hope the reviewers agree to go forward without the rescue data for the *unc-104* regeneration phenotype.

In the revision, we plan put the regeneration data into the supplemental data, and instead bring the synaptic overgrowth data from supplemental data (Figure 3—figure supplement 1) into the main text. The overgrowth data provide further support for the conclusion that Wnd signaling is activated in *unc-104* mutants.

Subsection “Activation of the Wnd signaling pathway in neurons is sufficient to impair presynaptic structure and synaptic transmission”: first paragraph, the authors state "ectopic activation of Wnd should mimic the unc-104 mutant phenotypes". But where is Wnd normally activated?

Please see our discussion above for reviewer 1’s point C. All of our data, combined with other data in the field for Wnd/DLK function in flies and worms, suggest its activity is restrained to very low levels in ‘normal’ wild type uninjured animals. Most previous work in the field has focused on Wnd/DLK’s role in injured neurons (when it is activated). There has previously been no supporting data (in flies or worms) to indicate that Wnd also functions in uninjured animals. In this present study we explored this question more deeply, through a careful analysis of *wnd loss-of-function* phenotypes during stages of embryonic development in Figure 7.

In the same section, second paragraph: The authors made a comment about the result in Figure 4 being differed from their previous study that used a hypomorphic allele of Wnd". They should perform the same analysis in parallel on both null and that hypomorphic allele.

Please refer to our discussion of this for reviewer 1’s point 2 above. Importantly, and similarly to the other points, this point is addressable and is also not essential for the main conclusions of this paper.

They talked about similarities of how loss of Hiw or of Unc-104 affect Wnd function. An important experiment should include a double mutant study of Hiw and unc-104.

We believe that this suggested experiment is to help in determining whether Hiw and Unc-104 regulate Wnd through the same or distinct pathways. However a consideration of the potential outcomes makes it unlikely that the experiment will yield new information. One would need to compare Wnd levels and localization in the *hiw; unc-104* double mutants to *hiw* and *unc-104* single mutants, which have very different phenotypes for both levels and localization. Because of the major transport defects in *unc-104* mutants, it is unlikely that Wnd localization and levels in the double mutant would resemble the *hiw* single mutant. Instead it is quite likely that one would observe increased levels of Wnd (due to loss of *hiw* function) AND an in increase in levels of Wnd in the cell body which occurs in the *unc-104* mutants. This kind of data would not help us distinguish whether Unc-104 and Hiw regulate Wnd through a common pathway or parallel pathways.

However to address the general question we could include some more discussion about potential relationships between Hiw and Unc-104. See our discussion for reviewer 2’s point 2. However this relationship is a discussion point that is not essential for the major conclusions of this paper.

So, in summary for this point we respectfully disagree that analysis of the *hiw; unc-104* double mutant is an important experiment for this paper. Instead, it would be reasonable to include the discussion to reviewer 2’s point 2 above in the published review and rebuttal section for *eLife*. This would be a great format for the field to see these additional points discussed, which are side-points for the main points of the paper.

[Editors' note: the author responses to the re-review follow.]

While we are happy to consider a revised manuscript, you will see that we have some serious concerns that need careful attention. We see merit in the author's rebuttal letter that the most concerns can be addressed with rewriting and some additional experiments and key issues addressed. The overall impact of the work, if these points are well-addressed, can be strong, since it provides a fundamentally new model for how unc-104 contributes to neuronal function, by activating an axonal injury pathway. However, the data and model also need to be explained more clearly to be digestible by a broad readership such as at eLife.

We appreciate the opportunity to address the points raised in the review. We have now provided the additional requested data, itemized below.

In addition, the questions raised in the first round of review helped us see how the writing and manuscript organization could be improved so that a broader audience could follow the narrative and the significance of our findings. To accomplish this we made changes in the order of data (Figure) presentation, and changes in text throughout the paper including the Abstract, Title, and Results. The accompanying ‘track changes’ version of the manuscript highlights the significant changes since the first version.

In response to the concerns raised about inconsistent results in their wnd loss-of-function analysis, the authors are dismissive, imploring us to "Note that our model is NOT that Wnd functions as a major restraint on these proteins during normal conditions or during development," yet their subheading describing these results is "Wnd restrains the expression of presynaptic proteins at early stages of synaptogenesis."

We appreciate this point, and apologize that the previous attempt to explain our thinking came off as dismissive. In our revision we have worked to communicate our interpretation of results more carefully.

The principal finding in the paper is that Wnd signaling accounts for synaptic defects in *unc-104* mutants by down-regulating the expression levels of SV and AZ protein components. However this is a pathological context, in *unc-104* mutants. We wondered whether Wnd signaling ever plays a similar role in wild type animals.

Previous studies have suggested that while Wnd/DLK signaling becomes activated in injured neurons, it is normally highly restrained in uninjured animals (Collins et al., 2006; Xiong et al., 2010), and, in contrast to its essential role in responses to axonal injury, roles for Wnd in developmental axonal outgrowth or synapse formation have not been previously described (Collins et al., 2006). Importantly, the function of the Wnd signaling pathway had not previously been examined at the critical developmental stages of synapse formation, due to the technical challenges of dissecting late-stage embryos (whose diameter is approximately the size of two human hairs). The late-stage embryo dissection skills that we mastered for this project allowed us to probe this question carefully with the results presented in Figure 8.

Our findings (in Figure 8) indicate that the effects of *wnd* mutations upon SV protein expression are indeed modest in comparison to the dramatic effects of *wnd* in *unc-104* mutants. However they are nevertheless interesting to report since they suggest that Wnd signaling also restrains synaptic protein levels during development in wild type, uninjured animals.

According to the modified text, we changed the subheading of this section to “Wnd delays the expression of SV proteins during early stages of synapse development”, and we have made large revisions in writing for this section.

In a second example, the authors dismiss concerns about their analysis of MiMIC-GFP instead of endogenous Wnd levels by simply stating that the Wnd antibody does not work well for immunohistochemistry without addressing the fact that they have previously published analogous experiments using their Wnd antibody for immunohistochemistry (Collins et al., 2006). The authors may want to pay particular attention to these points.

As the reviewers pointed out, the Wnd antibodies developed in (Collins et al., 2006) could successfully detect a robust and Wnd-specific signal by both immunohistochemistry (IHC) and western blot (WB) in *hiw* mutants, when Wnd protein is elevated. However in wild type as well as *unc-104* mutant animals, we were unable to detect a reproducible IHC signal above background in our current conditions using these antibodies. While some misfortune may be potentially attributed to poor antibody storage conditions, we consider the difference between *hiw* mutants (in which a signal can be detected) and *unc-104* mutants (where we failed to detect an IHC signal) to be meaningful.

For the additional reagents used to examine Wnd protein, we have added some additional explanation into the manuscript text. That section is quoted here:

“We then looked further at Wnd’s localization using available reagents. By immunocytochemistry (IHC) with anti-Wnd antibodies, Wnd protein remained below the detection threshold in both wild type and *unc-104* mutant animals (data not shown). […] This highly expressed transgene also revealed an increase in cell body localized Wnd in *unc-104* mutants (Figure 5).”

The authors should measure puc-LacZ expression in additional unc-104 mutants.

We have now included *puc*-lacZ measurements for additional *unc-104* alleles (*O3.1/170* and *bris/170*) in Figure 4. All of the mutations tested (including the RNAi-knockdown, shown previously) lead to significant induction of *puc*-lacZ expression (Figure 4).

Also, rescue of additional phenotypes aside from NMJ morphology and active zone apposition to PSD (e.g. lethality or synaptic function) by BSK and FOS would provide strong support for the model.

We have added new electrophysiology data which shows that inhibition of downstream Wnd signaling components Fos or Bsk mimics *wnd* mutations in rescue of the severely impaired spontaneous synaptic release (mEJP frequency) in *unc-104* mutants (Figure 3 and Figure 3—figure supplement 1). In addition, inhibition of Bsk rescued mEJP and EJP amplitude while inhibition of Fos rescued EJP amplitude and quantal content in the *unc-104* mutant background (Figure 3—figure supplement 1). These data add further support to the model that Wnd signaling mediates the synaptic transmission defects when Unc-104’s transport is disrupted.

We also included additional data that inhibition of Fos and Bsk (*JNK*) also mimic *wnd* mutations in the accumulation of SV and AZ proteins in the cell body of *unc-104* mutants (Figure 7—figure supplement 2). These data further indicate that the signaling cascade downstream of Wnd promotes the restraint of presynaptic protein levels in *unc-104* mutants.

However, I was not concerned that the Mimic tool showed an increase in Wnd while the western blot did not, since the Mimic tool accounts for changes in subcellular distribution – this point can be addressed better by the authors in the manuscript. However, it is absolutely essential that the MIMIC phenotype be quantified and use of the Kinase Dead explained in the text.

We have added quantification for the changes noted for MiMIC-*wnd*-GFP (Figure 5) and explanation of the GFP-*wnd-kinase dead* reagent in the text.

The effect of wnd on the timing and expression of presynaptic proteins needs to be much more clearly explained, as discussed in the rebuttal. As it stands this section is very distracting and confusing.

We have done significant re-writing for this section to more clearly explain the rationale, expectations and interpretations of the study of *wnd loss-of-function* phenotypes during synaptic development (Figure 8 and subsection “Wnd delays the expression of SV proteins during early stages of synapse development”).

In addition, based on the original round of review, we think that the data for Brp IHC in *wnd* mutants added an additional layer of confusion: in the initial manuscript (previously Figure 7—figure supplement 1) we saw that Brp levels were slightly reduced in *wnd* mutants, which contrasts with the enhanced levels we observed for SV associated proteins VGlut and Syt1 at early stages of synapse development. We argued in our appeal that this difference does not negate the model: since very little is known about how different AZ and SV proteins are regulated developmentally, there is no reason to predict or expect that Brp levels would be sufficiently high in stage 15-17 animals to be subject to restraint by Wnd signaling.

Since our observations with Brp did not negate the model but instead opened up new questions, we realized that it is more distracting than it is useful given our current state of knowledge. So we have decided to remove this data from the revised manuscript.

The authors should also clarify their model for how synaptic protein levels are regulated in the unc-104 mutants (though the authors do discuss the RT-PCR data and I agree with them that RT-PCR detects transcripts from all the cell types in all the tissues used for the mRNA prep, and does not account for specific defects in protein levels and localization at the neuronal cell bodies and synaptic terminals). The main point of the paper does not hinge on a particular model of synaptic protein level control, but the authors do need to clarify their explanations, either in the text or with more experiments.

As agreed by the reviewer, the major effect we observed by IHC on presynaptic protein expression occurs in motoneurons. Since the RT-PCR data was obtained from the entire larval CNS, this data is limited in sensitivity and utility for making conclusions about any changes in mRNA levels in motoneurons. Therefore we have removed this data (previously Figure 6—figure supplement 2).

And, as requested, we have now added further discussion for the question of HOW Wnd signaling may be regulating presynaptic protein levels, quoted below:

Quoting from revised text, in the Results:

”These observations, combined with the observations that Wnd activation reduces the levels of Brp and VGlut at NMJs, suggest a model in which activated Wnd signaling leads to a down-regulation of the expression levels of multiple pre-synaptic proteins in *unc-104* mutants. […] This, together with the involvement of transcription factor Fos in restraining VGlut and Brp buildup in cell bodies (Figure 7—figure supplement 2), implies that Wnd signaling may inhibit presynaptic protein expression at the transcriptional level.”

And in Discussion:

“How does Wnd signaling regulate presynaptic proteins? […] On the other hand, previous studies have separately linked Unc-104, Wnd signaling and AP-1 (Fos) to autophagy (Guo et al., 2012; Shen and Ganetzky, 2009; Stavoe et al., 2016). It will be interesting to determine whether translation and/or autophagy contribute to the regulation of presynaptic proteins by Wnd signaling.”

Wnd null mutants alone should be included in Figure 5 and Figure 6 for comparison in this specific assay.

Our revision includes these controls side-by-side in the figures for comparison. Due to our reorganization the previous Figure 5 (suppression of the *unc-104-null* phenotype) is now Figure 1, and the previous Figure 6 (dramatically elevated presynaptic protein in *unc-104;wnd-null* mutant embryos) is now Figure 7—figure supplement 1. For these phenotypes the *wnd* single mutants resemble wild type animals.

However, though phenotypes in transheterozygotes can be informative, lack of a dominant transheterozygote phenotype might just reflect gene dose requirements, and so we did not put as much weight on that particular experiment.

We have examined synaptic defects of transheterozygotes (*unc-104^bris/P350^;wnd^3/+^*) and found that heterozygous mutations in *wnd* fail to dominantly suppress the *unc-104* defects. Since this observation neither proves nor negates the overarching model, we decided not to include this data.

Electrophysiological analysis (and synaptic vesicle protein levels) in wnd overexpression lines would provide strong corroboration that increased Wnd explains functional defects in unc-104 mutants.

This was a great suggestion – this experiment provides more direct evidence than the analysis of hiw phenotypes (which is now moved to supplementary data). We provided these analysis in the revision, in Figure 6-I.

Quoting from subsection “Activation of the Wnd signaling pathway in neurons is sufficient to impair presynaptic structure and synaptic transmission”:

…“Indeed, over-expression of *wnd* alone in motoneurons resulted in cell-autonomous presynaptic defects that are comparable to *unc-104* mutants. […]Taken together with the rescue of *unc-104* synaptic defects by *wnd* mutations (Figure 1–Figure 3), these data indicate that activation of Wnd signaling leads to strong perturbations in presynaptic structure and function.”

Addressing the discrepancy with previous results (Collins 2006) on Wnd suppression of the hiw phenotype will be also be important.

First, it is important to clarify that our cumulative electrophysiological observations in the paper suggest that activation of the Wnd pathway does not strongly impair quantal content. Instead, the strongest and most salient defect caused by Wnd pathway activation is a reduction in spontaneous mEJP frequency. The frequency of detectable mEJP events was reduced in *unc-104^bris/P350^* and *hiw^ΔN^* mutants by ~70% and ~80%, respectively, and this defect was fully suppressed by mutations in *wnd* as well as downstream signaling components *bsk* and *fos* (Figure 3 and Figure 6—figure supplement 2). Also, mEJP frequency becomes strongly reduced when Wnd is ectopically expressed (Figure 6). We also observed modest but statistically significant reductions in mEJP amplitude mediated by Wnd (Figure 3, Figure 6 and Figure 6—figure supplement 2). These observations are consistent with the data reported for Wnd-mediated synaptic transmission defects in *hiw^ND8^* mutants in the 2006 study. The 2006 study did not quantify mEJP frequency, but the effect is apparent in the published traces, and is now quantified in our partial repeat of that data (Figure 6—figure supplement 2).

The only difference between our findings and the 2006 study was that the 2006 study showed that *wnd* mutations failed to suppress *hiw* defects in quantal content, while our data suggested that quantal content defects may be partially suppressed by *wnd* mutations. The two studies used different alleles of *hiw* and *wnd*, so to understand the basis for the differences, we carried out additional electrophysiology recordings for multiple allele combinations for side-by-side comparisons (Figure 6—figure supplement 2). Confirming/repeating the 2006 result, *hiw^ND8^; wnd^1/2^* double mutants showed the same defects in quantal content as *hiw^ND8^* mutants alone. The partial rescue by *wnd^[3]^/Df* mutations in *hiw^ΔN^*mutants is only of borderline significance (it was more significant when we used higher calcium bathing solution in our first submission). We noted that *hiw^ΔN^*, which is a true null allele (Wu et al., 2005) showed a slightly lower quantal content than *hiw ^ND8^*(Figure 6—figure supplement 2). This suggests the potential for a modest role of Wnd in the *hiw^ΔN^* quantal content phenotype, however further controls would be needed to fully address this possibility.

As discussed above, in contrast with the reduced mEJP frequency and mEJP size, altered quantal content is not a strong feature of Wnd pathway activation. So we have focused our conclusions in this study to the effect of Wnd pathway upon spontaneous rather than evoked neurotransmission.

On the other hand, experiments to address the cellular function of hiw would be nice but are not essential to the main interesting points of the paper. If the authors have the data or a more clear model of the role of hiw, it would be helpful to include, but we don't think it's necessary.

We have provided a more clear discussion of the differences between *unc-104* mutants and *hiw* mutants:

Quoting from subsection “Wnd protein is mislocalized to the cell body in unc-104 mutants”:

*… “*Mutations in *hiw* lead to strongly elevated levels of Wnd protein whose localization can be detected in neurites and at synapses (Collins et al., 2006). […] These observations suggest that the mechanism of Wnd signaling activation in *unc-104* mutants is unlikely to occur via Hiw.”

Interestingly, the activation of Wnd signaling in mutants for the spectroplakin shortstop (*shot)*(Valakh et al., 2013) bears more resemblance to *unc-104* mutants. (That is, in contrast to *hiw* mutants, and similarly to *unc-104* mutants, *shot* mutants also do not cause global increases in Wnd protein levels (Valakh et al., 2013)). Another recent study suggests that *shot, unc-104*, and *tau* mutants may influence a similar pathway (Voelzmann et al., 2016). This suggests a possibility that *unc-104* and *shot* mutants (which likely disrupt neuronal cytoskeleton) may share a similar mechanism of Wnd signaling activation.

The rescue of the axon regrowth phenotype are not critical for the main conclusions of the paper and could take a lot of time.

We appreciate this point. We observed enhanced axonal growth after injury both in *unc-104-hypomorph/null* mutants and in independent RNAi lines for *unc-104* (Figure 4). However showing rescue of this phenotype would require additional time.

[Editors' note: further revisions were requested prior to acceptance, as described below.]

Reviewer #1:

In their revised manuscript, the authors have been very responsive to reviewer suggestions. The logical flow of the manuscript is significantly improved and experimental concerns directly addressed.I have one question regarding the addition of wnd null data to Figure 1 and Figure 7—figure supplement 1. Is it still accurate to state that "Control and Experimental animals were always dissected, fixed and stained in the same dish/slide and imaged in parallel using identical confocal settings" (Experimental Procedures,), or were the new data obtained in separate experiments and normalized to wild type for incorporation with previously obtained data?

Good point – for the embryo data we needed to dissect and fix independently but these were fixed for the same time points and imaged in parallel with identical settings. We have modified the quoted text to be more accurate by replacing “same dish/slide” with “same condition”. See subsection “Immunocytochemistry”.